# Meningeal lymphatics clear erythrocytes that arise from subarachnoid hemorrhage

Jinman Chen [1,2,3,4], Linmei Wang [5], Hao Xu [1,2,4], Lianping Xing [6], Zixin Zhuang [1,2,3,4], Yangkang Zheng [1,2,4], Xuefei Li [1,2,4], Chinyun Wang [1,2,7], Shaohua Chen [1,2,4], Zibin Guo [8], Qianqian Liang [1,2,4 ✉] & Yongjun Wang [1,2,3,4 ✉]

Extravasated erythrocytes in cerebrospinal fluid (CSF) critically contribute to the pathogenesis of subarachnoid hemorrhage (SAH). Meningeal lymphatics have been reported to drain macromolecules and immune cells from CSF into cervical lymph nodes (CLNs). However, whether meningeal lymphatics are involved in clearing extravasated erythrocytes in CSF after SAH remains unclear. Here we show that a markedly higher number of erythrocytes are accumulated in the lymphatics of CLNs and meningeal lymphatics after SAH. When the meningeal lymphatics are depleted in a mouse model of SAH, the degree of erythrocyte aggregation in CLNs is significantly lower, while the associated neuroinflammation and the neurologic deficits are dramatically exacerbated. In addition, during SAH lymph flow is increased but without significant lymphangiogenesis and lymphangiectasia. Taken together, this work demonstrates that the meningeal lymphatics drain extravasated erythrocytes from CSF into CLNs after SAH, while suggesting that modulating this draining may offer therapeutic approaches to alleviate SAH severity.

[1] Longhua Hospital, Shanghai University of Traditional Chinese Medicine, 725 Wan-Ping South Road, 200032 Shanghai, China. [2] Spine Institute, Shanghai University of Traditional Chinese Medicine, 725 Wan-Ping South Road, 200032 Shanghai, China. [3] School of Rehabilitation Science, Shanghai University of Traditional Chinese Medicine, 1200 Cailun Road, 201203 Shanghai, China. [4] Key Laboratory of Theory and Therapy of Muscles and Bones, Ministry of Education (Shanghai University of Traditional Chinese Medicine), 1200 Cailun Road, 201203 Shanghai, China. [5] Department of Anatomy, School of Basic Medicine, Shanghai University of Traditional Chinese Medicine, 1200 Cailun Road, 201203 Shanghai, China. [6] Department of Pathology and Laboratory Medicine and Center for Musculoskeletal Research, University of Rochester Medical Center, 601 Elmwood Avenue, Rochester, NY 14642, USA. [7] The International Education College, Nanjing University of Chinese Medicine, 138 Xianlin Road, 210029 Nanjing, China. [8] The Fourth Clinical Medical College, Guangzhou University of Traditional Chinese Medicine, 232 Huandong Road, 510006 Guangdong, China. ✉email: liangqianqian@shutcm.edu.cn; YJwang8888@126.com

A s estimated by Global Burden of Diseases (GBD 2016), stroke is the second leading cause of death[1]. Subarachnoid hemorrhage (SAH) contributes to only 5% of the cases of stroke[2], but it occurs at a fairly young age and is associated with a high degree of mortality (in the range of 50%)[3]. Thus SAH is a large burden on society. The progression of SAH includes early brain injury that appears the first 3 days after injury, followed by delayed cerebral ischemia (DCI) 3–4 days after injury, which reaches the highest incidence and severity 6–8 days after SAH[4]. As vasospasm, microthrombosis, and neuroinflammation contribute greatly to the DCI and clinical prognosis, pharmacological approaches to treat this pathology have been focused on preventing vasospasm, microclot formation, and anti-inflammation for decades; however, to date, few agents have been found to exert beneficial effects on patient outcomes[5,6]. The pathologies of SAH are caused by the extravasated blood presenting in the subarachnoid space (SAS). Thus a better understanding of extravasated blood clearance may help in the development of effective therapeutic approaches. When a SAH attack occurs, blood releases into the SAS, leading immediately to clot formation, which disappear within 2–3 days[7,8]. To date, clot lysis and phagocytosis by macrophages and neutrophils were considered as ways to clear the extravasated blood and the subsequent clots in the SAS[9,10]. However, the mechanism(s) by which extravasated blood is cleared is still unclear.

Recently, lymphatic vessels have been rediscovered and characterized in the meninges surrounding the central nervous system[11,12]. These lymphatics are responsible for the drainage of cerebrospinal fluid (CSF) macromolecules and immune cells to the cervical lymph nodes (CLNs)[13,14]. Meningeal lymphatics aid in the clearance of amyloid-beta in aged mice and transgenic mouse models of Alzheimer's disease, and the augmentation of meningeal lymphatic function alleviates age-associated cognitive impairment[13,15–17]. But in an experimental autoimmune encephalomyelitis (EAE) mouse model the ablation of meningeal lymphatics or inhibition of vascular endothelial growth factor receptor 3 (VEGFR3) diminished the EAE severity and the inflammatory response of brain-reactive T cells[14,18]. Thus the meningeal lymphatics may play different roles in different neurological diseases. Embryonic mesenteric lymphatic vessels have recently been shown to clear extravascular red blood cells (RBCs) leaking from the adjacent developmental vascular remodeling, suggesting that the lymphatic system has the potential to clear the erythrocytes in interstitial fluid[19]. But, as noted above, while meningeal lymphatics drain immune cells and macromolecules from CSF, it is still unclear whether they clear the extravasated blood after SAH.

Here we report that the extravasated blood is aggregated in CLNs, and the erythrocytes are accumulated in the lymphatics of CLNs inside and around meningeal lymphatic vessels for 4 h after SAH. We use CFSE (5-(and 6)-carboxyfluorescein diacetate succinimidyl ester) to label erythrocytes ex vivo and inject them into the cisterna magna and demonstrate that the labeled erythrocytes are also evident in the lymphatics of CLNs and meningeal lymphatics. When we ablate meningeal lymphatics by photoconverted visudyne, RBC drainage into CLNs is significantly reduced, while the neuroinflammation and the neurological deficits associated with SAH are exacerbated. We further block VEGFR3 and also observe a worse brain injury during SAH. We examine the structure and functional characteristics of meningeal lymphatics during neuroinflammation and find the lymph flow is augmented but lymphangiogenesis and expansion are not pronounced. Our findings demonstrate that meningeal lymphatics participate in the drainage of RBCs into CLNs during the very early stage of SAH and may be a potent target in the treatment of this pathology.

## Results

**RBCs in CSF are drained by the meningeal lymphatics to CLNs.** Tracers, proteins, and labeled T cells injected into the brain or CSF are found in CLNs, indicating the drainage function by meningeal lymphatics[14,20]. To explore whether the erythrocytes released into the CSF, a condition called SAH, are drained to CLNs via the meningeal lymphatics, we injected autologous blood into the cisterna magna of mouse. Four hours later, we observed that the deep CLNs (dCLNs) and mandibular LNs were infused with blood, while no evidence of blood was observed in the saline-injected and control groups (Fig. 1a, Supplementary Fig. 1a, b). The data also suggest that the superficial parotid LNs, though located in the superficial anterior neck, did not drain the CSF erythrocytes, and there is no drainage of the axillary, the brachial, and the tracheobronchal LNs (Supplementary Fig. 1a). Next, we used anti-Ter119 antibody to label erythrocytes and anti-Lyve-1 antibody to visualize the lymphatics of CLNs. The number of Ter119-positive erythrocytes in the Lyve-1-positive lymphatic sinus was significantly greater in the SAH group when compared with the saline and control groups for both types of CLNs (for the comparison of dCLNs, $P = 0.002$, Con vs SAH, $P = 0.0063$, saline vs SAH, Fig. 1b, c; Supplementary Fig. 1c, d). We found that the Ter119-positive cells were distributed mainly in the Lyve-1-positive lymphatic sinus of CLNs.

To determine whether the erythrocytes are drained to CLNs via the meningeal lymphatics, we isolated the meninges and stained the lymphatic vessels with antibodies. Erythrocytes were observed to accumulate around and inside of meningeal lymphatics at 4 h post-induction of SAH, while erythrocytes were not seen significantly entering into the lymphatics of the control and saline-injected groups ($P < 0.0001$, Con vs SAH; $P < 0.0001$, saline vs SAH; Fig. 1d, e). The corresponding orthogonal view (Supplementary Fig. 2) showed the erythrocytes co-localizing with Lyve-1-positive lymphatic endothelial cells, suggesting that these cells were drained via the meningeal lymphatic vessels. The identification of meningeal lymphatics was further confirmed by co-labeling them with other classical lymphatic endothelial cell markers including podoplanin (PDPN) and Prox1 with lyve-1, and the erythrocytes also showed accumulation into meningeal lymphatics (Fig. 1f). Morphologically intact erythrocytes and clusters of degraded membranes were seen at the same time in the lymphatics of CLNs and meninges (Fig. 1b, d), though it is not clear whether the erythrocytes were broken down before transportation by the lymphatics or after. But the data suggest that at least some erythrocytes were drained into CLNs before degradation.

To further confirm that the erythrocytes in meningeal lymphatics and CLNs were drained from exogenous injection, we labeled erythrocytes with CFSE in vitro and then injected them into the cisterna magna. Four hours later, the labeled erythrocytes also showed similar accumulation in the meningeal lymphatics ($P < 0.0001$, Fig. 2a, b), dCLNs ($P = 0.0006$, Fig. 2c, e), and mandibular LNs ($P = 0.0207$, Fig. 2d, e), while the saline-treated group did not show any evidence of CFSE-labeled erythrocytes.

**Ablative meningeal lymphatics block RBC drainage to CLNs.** To further verify whether meningeal lymphatic vessels serve as the route for the drainage of erythrocytes from the CSF into the CLNs, we ablated meningeal lymphatic vessels by injecting visudyne into the cisterna magna and photoconverted it by laser light. After 7 days of ablation, lymphatic coverage of the transverse sinus ($P < 0.0001$, Con vs Laser + Visudyne; $P < 0.0001$, Laser vs Laser + Visudyne; $P < 0.0001$, Visudyne vs Laser + Visudyne; Fig. 3a, b) and the superior sagittal sinus were notably

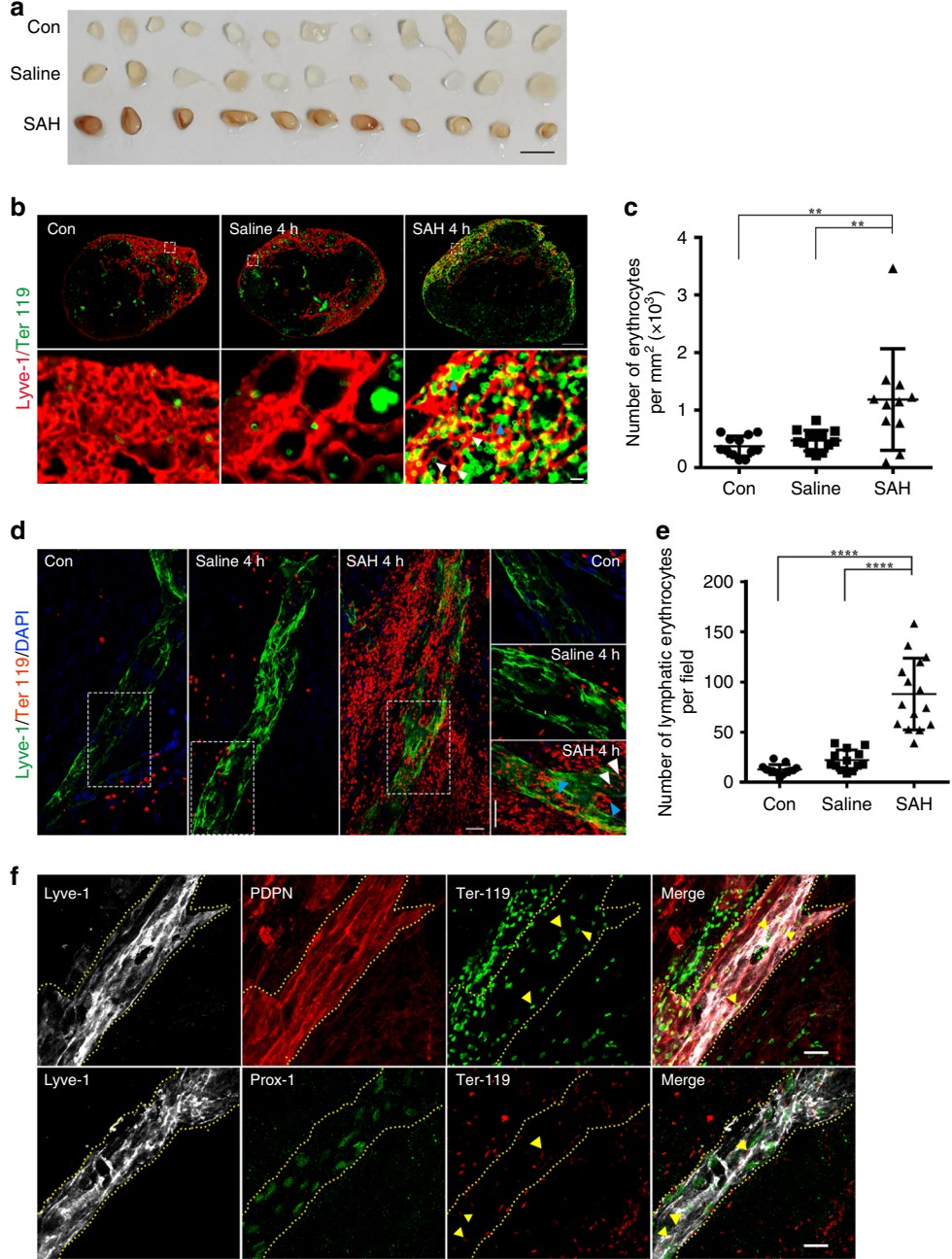

**Fig. 1 The erythrocytes in the cerebrospinal fluid after subarachnoid hemorrhage (SAH) were drained into deep cervical lymph nodes (dCLNs) via meningeal lymphatics. a** Isolated dCLNs from mice 4 h after autologous blood (SAH) or saline injected into the cisterna magna (i.c.m.) or blank control group (Con). Scale bar, 2 mm. **b** Representative images of dCLNs isolated 4 h post-induction of the SAH, saline, and Con groups, stained for Lyve-1 (to image lymphatics, red) or Ter119 (to image erythrocytes, green). Regions of interest in the top images are shown below. White arrows, morphologically intact erythrocytes. Blue arrows, clusters of degraded membrane. Scale bars 200 μm (for top images) and 10 μm (for bottom images). $n = 3$ independent experiments. **c** The number of erythrocyte per mm$^2$ in the Lyve-1-positive lymphatic sinus of dCLNs from the SAH, saline, and Con groups. Con/Saline; $n = 12$, SAH; $n = 11$ mice, pooled from 3 independent experiments. $P$(Con vs SAH) = 0.002; $P$(Saline vs SAH) = 0.0063. **d** Representative image of the meningeal lymphatics (green) on transverse sinus (TS) draining the erythrocytes (red) after 4 h of SAH, compared with the Con and saline groups. Regions of interest in the left images are shown on the right. White arrows, morphologically intact erythrocytes. Blue arrows, clusters of degraded membrane. Scale bar, 20 μm (the left and right images). $n = 3$ independent experiments. **e** Quantification of the number of erythrocytes per field accumulated into lymphatics on TS. Con/Saline; $n = 13$, SAH; $n = 15$ biologically independent animals, pooled from 3 independent experiments. $P$(Con vs SAH) < 0.0001; $P$(Saline vs SAH) < 0.0001. **f** The expression of classical lymphatic endothelial cell markers by the meningeal lymphatic vessels co-stained with erythrocytes. Representative images of podoplanin (PDPN, red) and Lyve-1 (gray) expressing vessels co-localized with erythrocytes (green) (top images). Representative images of Prox1 (green) and Lyve-1 (gray) expressing vessels co-localized with erythrocytes (red) (bottom images). Yellow arrow, erythrocytes inside the meningeal lymphatic vessels. Scale bar, 20 μm. $n = 2$ independent experiments. All data are presented as mean values ± SD; one-way ANOVA with Turkey's multiple-comparison test. **$P$ < 0.01, ****$P$ < 0.0001. Source data are provided as a Source data file.

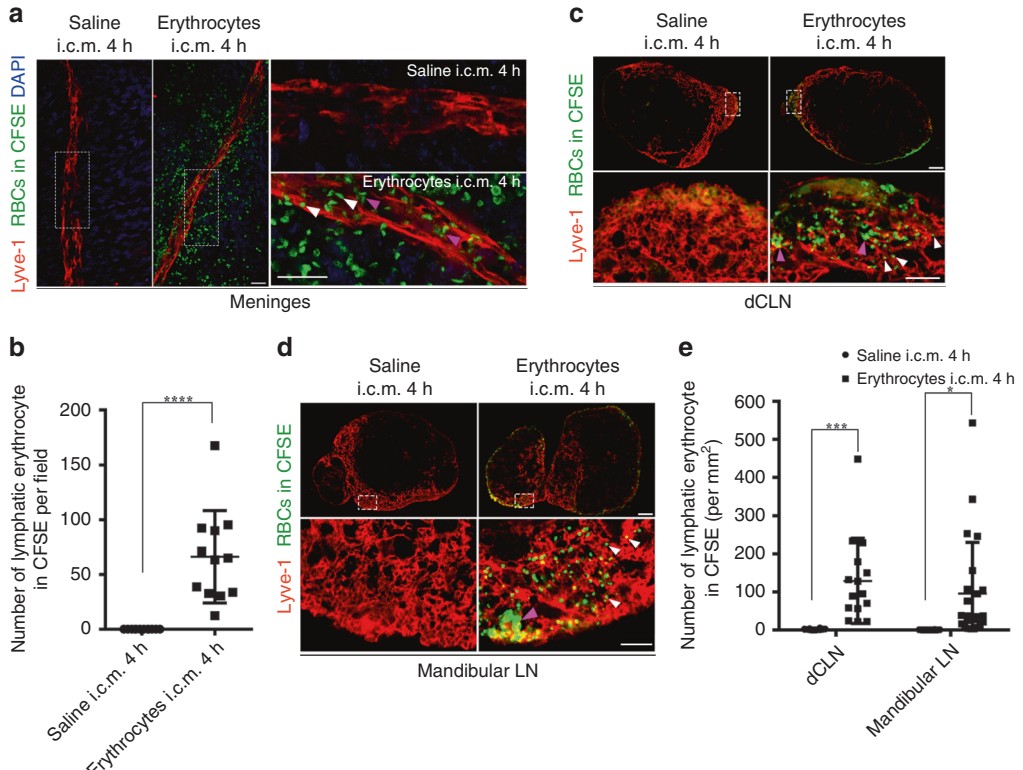

**Fig. 2 Meningeal lymphatics drained erythrocytes labeled in vitro to cervical lymph nodes (CLNs).** Erythrocytes (red blood cells (RBCs)) were labeled by CFSE (5-(and 6)-carboxyfluorescein diacetate succinimidyl ester) in vitro and then injected into cistern magna. **a** Representative images of meningeal lymphatics draining erythrocytes in CFSE at 4 h postinjection. White arrows, morphologically intact erythrocytes. Purple arrows, degraded membrane. Regions of interest in the left images are shown on the right. Scale bar, 20 μm (the left and right images). $n = 2$ independent experiments. **b** Quantification of the number of CFSE-labeled erythrocytes per field in meningeal lymphatics on TS ($n = 12$ mice per group, pooled from 2 independent experiments. $P < 0.0001$). **c** Representative images of CFSE-labeled erythrocytes in the Lyve-1-positvie lymphatic sinus of dCLNs of mice at 4 h postinjection. Regions of interest in the top images are shown below. White arrows, morphologically intact erythrocytes. Purple arrows, clusters of degraded membrane. Scale bars, 200 μm (the top images) and 50 μm (the bottom images). **d** Representative images of CFSE-labeled erythrocytes in the Lyve-1-positive lymphatic sinus of mandibular LNs of mice at 4 h postinjection. White arrows, morphologically intact erythrocytes. Regions of interest in the top images are shown below. Purple arrows, clusters of degraded membrane. Scale bars, 200 μm (the top images) and 50 μm (the bottom images). $n = 3$ independent experiments. **e** Quantification of the density of CFSE-labeled erythrocytes per mm$^2$ in Lyve-1-positive lymphatic sinus of dCLNs and mandibular LNs at 4 h postinjection (dCLN, Saline; $n = 12$, Erythrocytes in CFSE; $n = 15$ independent nodes, $P = 0.0006$; Mandibular LN, Saline; $n = 12$, Erythrocytes in CFSE; $n = 23$ nodes, pooled from 3 independent experiments. $P = 0.0207$). All data are presented as mean values ± SD; two-tailed unpaired Student's $t$ test. $^*P < 0.05$, $^{***}P < 0.001$, $^{****}P < 0.0001$. Source data are provided as a Source data file.

lower ($P < 0.0001$, Con vs Laser+Visudyne; $P = 0.0007$, Laser vs Laser + Visudyne; $P = 0.0003$, Visudyne vs Laser + Visudyne; Fig. 3c). No difference in cerebral blood flow was observed between control mice and the Laser + Visudyne group (Supplementary Fig. 3a, b). The coverage of blood vasculature on the transverse sinus and superior sagittal sinus were not altered in the Visudyne-photoconverted group compared with Laser only or Visudyne only group (Supplementary Fig. 3c, d).

On the seventh day after the ablation of the meningeal lymphatics, we induced SAH. After 4 h of SAH, blood infusion into the dCLNs and mandibular LNs were significantly lower in the mice with impaired meningeal lymphatics compared to the Laser only and the Visudyne only groups (Fig. 3d, Supplementary Fig. 3e, f). And the number of Ter119-labeled erythrocytes was markedly lower in the dCLNs ($P = 0.0013$, L + SAH vs L + V + SAH; $P = 0.0060$, V + SAH vs L + V + SAH; Fig. 3e, f) and mandibular LNs (Supplementary Fig. 3g, h) of the lymphatic-ablation group. In general, blood is released into the SAS when SAH occurs and is cleared within 2–3 days[7,8]. But we found that clot clearance was blocked when the meningeal lymphatics were

ablated as clots persisted in the brain until 7 days after the induction of SAH in the Laser + Visudyne group (Fig. 3g).

**Ablation of meningeal lymphatics worsens SAH severity.** Neuroinflammation is prominent in SAH, leading to cerebral cell damage and vasospasm[21]. Microglia are resident brain macrophages, can be activated in SAH, and can display either classical pro-inflammatory phenotype or alternative anti-inflammatory phenotype polarization[22]. We hypothesized that, after ablation of meningeal lymphatics, the microglia activation would be worsened owing to the prolonged exposure to erythrocytes degradants. We thus performed whole-brain flow cytometry to determine the ratio of CD16/32-posititive pro-inflammatory microglia and CD206-positive anti-inflammatory microglia. For these measurements, CD11b+CD45$^{low}$ cell populations were gated as myeloid lineage cells including microglia, in which the CD16/32 was used as a marker for cells with pro-inflammatory phenotype and CD206 was used as a marker for cells with anti-inflammatory phenotype. The gating strategy is shown in Fig. 4a. Compared with SAH mice with intact lymphatics (SAH only, L + SAH and V + SAH group), SAH

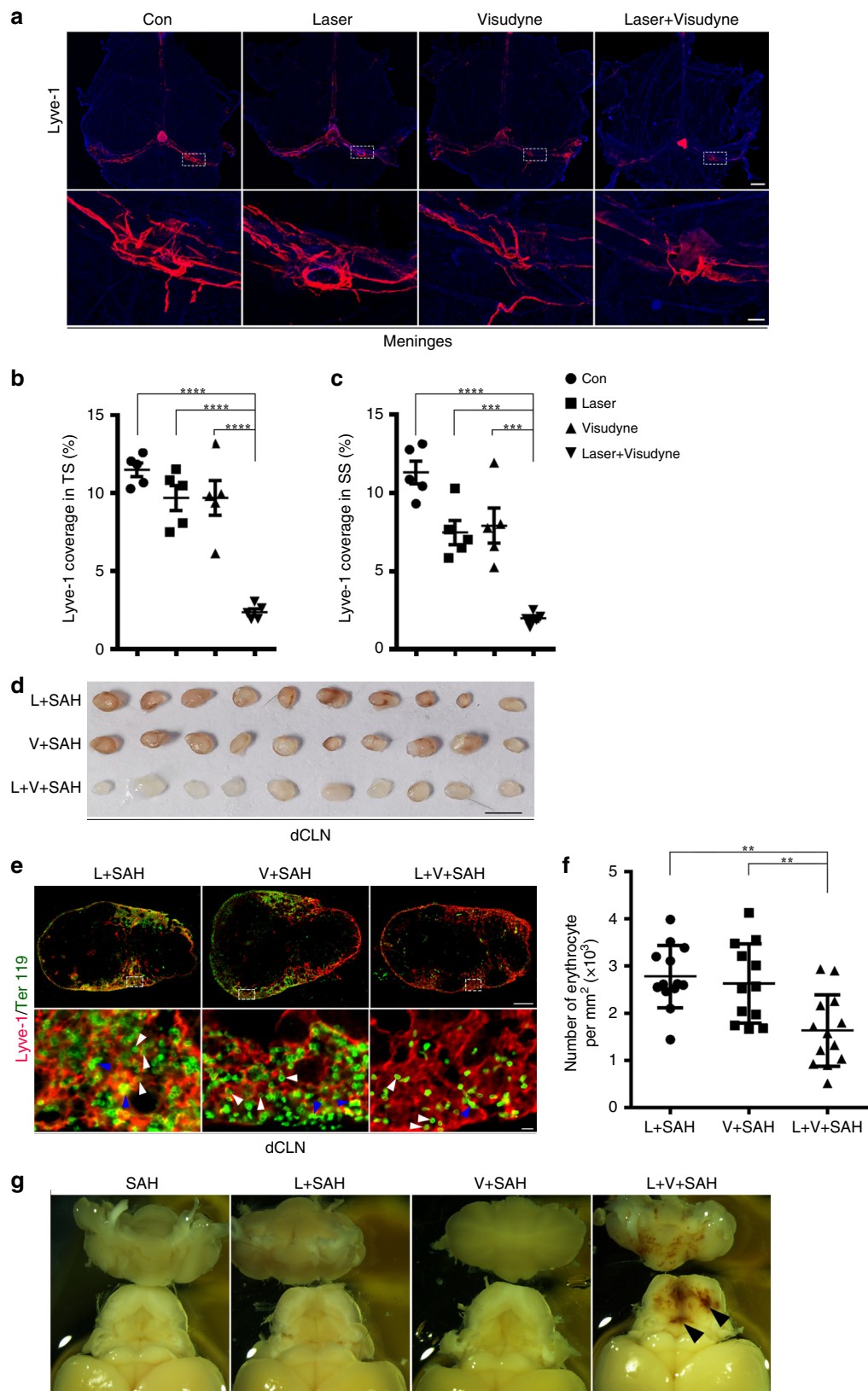

mice with lymphatic ablation (L + V + SAH group) showed a significantly greater percentage of CD16/32$^+$CD206$^-$ microglia ($P < 0.0001$ SAH/L + SAH/V + SAH vs L + V + SAH, Fig. 4b, c) and a lower percentage of CD206$^+$CD16/32$^-$ microglia ($P = 0.0146$, SAH vs L + V + SAH; $P < 0.0001$, L + SAH vs L + V +

SAH Fig. 4b, d). When compared with the Laser + Visudyne group, the degree of activated microglia polarized to a pro-inflammatory phenotype was markedly greater after SAH as evidenced by an increased ratio of CD16/32$^+$CD206$^-$ subsets and decreased ratio of CD206$^+$CD16/32$^-$ cells. ($P < 0.0001$, L + V vs L + V + SAH for

**Fig. 3 Ablation of meningeal lymphatic vessels blocks drainage of the erythrocytes into dCLNs. a** Representative images of meningeal lymphatics (labeled by Lyve-1, red) in the Con, Laser only, Visudyne only, and Laser plus Visudyne groups. Regions of interest in the top images are shown below. Scale bars, 1 mm (top images) and 200 μm (bottom images). **b** Quantification of the meningeal lymphatics coverage on the TS ($n = 5$ per group, $P$(Con vs Laser + Visudyne) < 0.0001, $P$(Laser vs Laser + Visudyne) < 0.0001, $P$(Visudyne vs Laser + Visudyne) < 0.0001) and **c** superior sagittal sinus (SSS) from each group of mice ($n = 5$ per group, $P$(Con vs Laser + Visudyne) < 0.0001, $P$(Laser vs Laser + Visudyne) = 0.0007, $P$(Visudyne vs Laser + Visudyne) = 0.0003). **d** Isolated dCLNs from mice 4 h after induction of SAH in the Laser (L + SAH), Visudyne (V + SAH), and Laser plus Visudyne (L + V + SAH) groups. Scale bar, 2 mm. **e** Representative images of dCLNs isolated 4 h post-induction of SAH and stained for the lymphatics (Lyve-1, red) and erythrocytes (Ter119, green) in the 3 groups of mice indicated. White arrows, morphologically intact erythrocytes. Blue arrows, clusters of degraded membrane. Regions of interest in the top images are shown below. Scale bars, 200 μm (the top images) and 10 μm (the bottom images). $n = 2$ independent experiments. **f** The number of erythrocyte per mm$^2$ in Lyve-1-positive lymphatic sinus of dCLNs 4 h after SAH induction (L + SAH/L + V + SAH; $n = 13$, V + SAH; $n = 12$, pooled from 2 independent experiments. $P$(L + SAH vs L + V + SAH) = 0.0013, $P$(V + SAH vs L + V + SAH) = 0.006). **g** Representative images of the brain after 7 days of induction of SAH in the groups indicated. Arrows, blood clot in the brain. Scale bar, 1 mm. All data are presented as mean values ± SD, one-way ANOVA with Turkey's multiple-comparison test, **$P < 0.01$, ***$P < 0.001$, ****$P < 0.0001$. Source data are provided as a Source data file.

CD16/32$^+$CD206$^-$ and $P = 0.0086$, L + V vs L + V + SAH for CD206$^+$CD16/32$^-$, Fig. 4c, d). The percentages of CD16/32 and CD206 double-positive cells, an intermediate state of polarization, and difference were only observed in the comparison between V + SAH and L + V + SAH groups ($P = 0.025$, Fig. 4e).

To assess the impact of the ablation of meningeal lymphatics on neurological function of the mice with SAH, we applied the open field test to evaluate the exploratory behavior of mice. SAH mice in the lymphatic-ablated group showed a significant decrease in the percentage of time spent in the center and the number of entrances into the center compared to the non-ablated SAH mice ($P = 0.0194$, SAH vs L + V + SAH, $P = 0.0053$, L + SAH vs L + V + SAH; $P = 0.009$, V + SAH vs L + V + SAH, Fig. 4f, $P < 0.0001$, SAH vs L + V + SAH; $P = 0.0001$, L + SAH vs L + V + SAH; $P = 0.0018$, V + SAH vs L + V + SAH; Fig. 4g). We also used a Y-maze test to evaluate the short-term working memory of the SAH model. We found that the SAH mice with ablated meningeal lymphatics were more vulnerable than the mice with intact lymphatics, as demonstrated by the percentage of time spent in the novel arm (NA) ($P = 0.0006$, SAH vs L + V + SAH; $P = 0.003$, L + SAH vs L + V + SAH, Fig. 4h) and the number of entrances into the NA ($P < 0.0001$, SAH vs L + V + SAH; $P < 0.0001$, L + SAH vs L + V + SAH; $P = 0.0338$, V + SAH vs L + V + SAH, Fig. 4i). When compared with the mice only treated with photoconverted visudyne, the mice accompanied with SAH showed worse performance in both the open field ($P = 0.0138$, time spent in the center; $P = 0.0141$, number of entries into the center; L + V vs L + V + SAH, Fig. 4f, g) and the Y-maze test ($P = 0.0198$, time spent in the NA arm; $P = 0.0303$, number of entries into the NA arm, Fig. 4h, i).

**Inhibition of VEGFR3 exacerbates SAH pathology.** Meningeal lymphatics maintain the potential to grow or regress in adults[13,18]. To further confirm the lack of integrity of meningeal lymphatics resulting in unfavorable outcomes in SAH, we inhibited VEGFR3, a tyrosine kinase receptor that promotes lymphangiogenesis, with MAZ51, a chemical inhibitor of VEGFR3 with proven effectiveness in inhibiting lymphangiogenesis[18]. MAZ51 was given intraperitoneally for 30 days (Fig. 5a). As shown in Fig. 5b, meningeal lymphatics underwent regression after treatment with MAZ51, as evidenced by a reduction in the lyve-1-positive area in both the transverse and superior sagittal sinuses (Fig. 5c). We observed no change in cerebral blood flow (Supplementary Fig. 3a, b) and of CD31-positive blood vasculature area of the sinuses (Supplementary Fig. 3i–j).

On the 30th day after treatment, vehicle- and MAZ51-treated mice were injected autologous blood or saline. Microglia

activation was also detected by flow cytometry (the general gating strategy is shown in Fig. 5d). As our data show, the percentage of CD16/32$^+$CD206$^-$ pro-inflammatory microglia was increased after SAH induction ($P = 0.0139$, Vehicle + Sham vs Vehicle + SAH; $P < 0.0001$, MAZ51 + Sham vs MAZ51 + SAH, Fig. 5e, f), but the proportion of activated microglia polarized to CD16/32$^+$CD206$^-$ phenotype were greater in the mice with meningeal lymphatic regression ($P < 0.0001$, Vehicle + SAH vs MAZ51 + SAH, Fig. 5f). The percentage of CD206$^+$CD16/32$^-$ anti-inflammatory microglia was not markedly affected by SAH or MAZ51 induction but significantly reduced after SAH + MAZ51 treatment ($P = 0.0233$, MAZ51 + SAH vs Vehicle + Sham) (Fig. 5g). For the cell population characterized by CD16/32$^+$CD206$^+$, no statistical significance was observed (Fig. 5h). These results were consistent with the above findings, indicating that the deterioration of the meningeal lymphatic vessels leads to further aggravation of neuroinflammation caused by SAH.

To validate the regression of meningeal lymphatics resulting in worse neurological outcomes after SAH, we performed the behavioral tests on MAZ51-induced lymphatic regression mice. The exploratory behavior and short-term working memory were assessed by open field test and Y-maze test, respectively. Mice with degenerated meningeal lymphatics spent less time in the center after SAH ($P = 0.0111$, Vehicle + SAH vs MAZ51 + SAH, Fig. 5i) but did not show any difference in the number of entries into the center (Fig. 5j). The deficits of short-term working memory after SAH also worsened in mice treated with MAZ51, represented by less time spent in the NA ($P = 0.0001$, MAZ51 + Sham vs MAZ51 + SAH, Fig. 5k) and the number of entries into the NA ($P = 0.0037$, MAZ51 + Sham vs MAZ51 + SAH; $P = 0.0221$, Vehicle + SAH vs MAZ51 + SAH, Fig. 5l).

**Characterization of meningeal lymphatics in SAH.** It has been reported that during acute inflammation, such as in intestinal inflammation, colitis, endocarditis, and rheumatoid arthritis, lymphatic flow and lymphangiogenesis are increased in the local peripheral lymphatic vessels, which is a mechanism for the body to reduce inflammation and edema[23–28]. To explore whether the neuroinflammation following SAH affects the function of the meningeal lymphatic vessels, we injected AF$^{488}$-conjugated anti-Lyve-1 or fluorescent microbeads into the cisterna magna at day 7 after induction of SAH or saline injection. After 30 min, meninges were harvested and stained for Lyve-1 using AF$^{555}$-conjugated secondary antibody. The percentage of meningeal lymphatics labeled by AF$^{488}$-anti-Lyve-1 antibody (intracisterna magna (i.c.m.)) indicates the speed of meningeal lymphatic flow. A

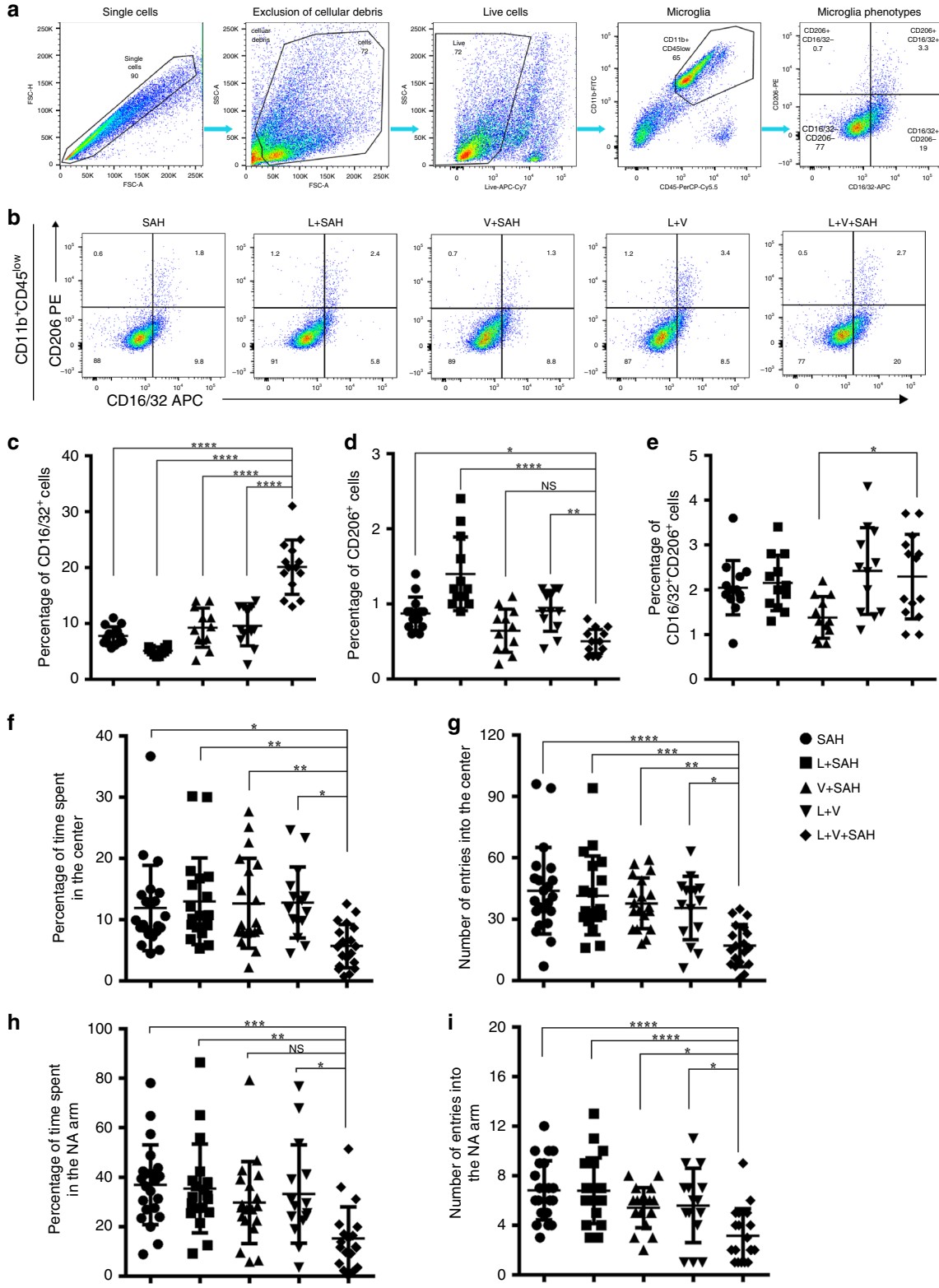

significantly greater percentage of meningeal lymphatics labeled by AF[488] anti-Lyve-1 (i.c.m.) was seen in the SAH group vs the controls (P = 0.0122, Con vs SAH; P = 0.0303, Saline vs SAH; Fig. 6a–c), and all the lymphatics labeled by injected antibody were seen on the transverse sinus. A greater percentage of lymphatics labeled by AF[488] anti-Lyve-1 (i.c.m.) also could be seen in dCLNs in the SAH group vs controls (P = 0.0183, Con vs SAH;

P = 0.0181, saline vs SAH; Fig. 6d, e). Similarly, 2 h after fluorescent microbead injection, the bead coverage in dCLNs was significantly higher in the SAH group vs the controls (P = 0.0004, Con vs SAH; P = 0.0251, saline vs SAH; Fig. 6f–h). In addition, the higher contraction frequency of mandibular afferent lymphatic vessels was detected in the SAH group than that in the saline group (P = 0.0261, Fig. 6i, j).

**Fig. 4 Ablation of meningeal lymphatics aggravates the neuroinflammatory response, the defects of exploratory behavior, and short-term working memory in SAH. a–d** Impaired meningeal lymphatics worsened the microglia activation in SAH. **a** Representative FACS plots showing gating strategy we used in flow cytometric analysis. Cell populations in the right dot plots defined as CD11b$^+$CD45$^{low}$ (microglia) were gated for further analysis. **b** Representative dot plots showing the ratio of CD16/32-positive subsets and CD206-positive subsets from different groups. **c** Quantification of CD16/32$^+$CD206$^-$ subsets of CD11b$^+$CD45$^{low}$ populations ($P$(SAH vs L + V + SAH) < 0.0001, $P$(L + SAH vs L + V + SAH) < 0.0001, $P$(V + SAH vs L + V + SAH) < 0.0001, $P$(L + V vs L + V + SAH) < 0.0001), **d** CD206$^+$CD16/32$^-$ subsets of CD11b$^+$CD45$^{low}$ populations ($P$(SAH vs L + V + SAH) = 0.0146, $P$(L + SAH vs L + V + SAH) < 0.0001, $P$(L + V vs L + V + SAH) = 0.0086), and **e** CD16/32$^+$CD206$^+$ subsets of CD11b$^+$CD45$^{low}$ populations in different groups ($P$(V + SAH vs L +V + SAH) = 0.025). SAH/L + V + SAH; $n = 14$, L + SAH/V + SAH/L + V; $n = 12$, pooled from 2 independent experiments. **f–i** Ablation of meningeal lymphatics worsens the exploratory behavior and short-term working memory associated with SAH. **f** Percentage of time spent in the center area in the open field test ($P$(SAH vs L + V + SAH) = 0.0194, $P$(L + SAH vs L + V + SAH) = 0.0053, $P$(V + SAH vs L + V + SAH) = 0.009, $P$(L + V vs L + V + SAH) = 0.0138), **g** the number of entries into the center area in the open field test ($P$(SAH vs L + V + SAH) < 0.0001, $P$(L + SAH vs L + V + SAH) = 0.0001, $P$(V + SAH vs L + V + SAH) = 0.0018, $P$(L + V vs L + V + SAH) = 0.0141), **h** percentage of time spent in the novel arm (NA) in the Y-maze test ($P$(SAH vs L + V + SAH) = 0.0006, $P$(L + SAH vs L + V + SAH) = 0.003, $P$(L + V vs L + V + SAH) = 0.0198), and **i** the number of entries into the NA in the Y-maze test in the groups of mice indicated ($P$(SAH vs L + V + SAH) < 0.0001, $P$(L + SAH vs L + V + SAH) < 0.0001, $P$(V + SAH vs L + V + SAH) = 0.0338, $P$(L + V vs L + V + SAH) = 0.0303). SAH; $n = 22$, L + SAH/V + SAH/L + V + SAH; $n = 19$, L + V; $n = 15$ mice, pooled from 2 independent experiments. All data are presented as mean values ± SD, one-way ANOVA with Turkey's multiple-comparison test, *$P$ < 0.05, **$P$ < 0.01, ***$P$ < 0.001, ****$P$ < 0.0001. NS, not significant. APC allophycocyanin, PE R-phycoerythrin, FITC fluorescein isothiocyanate. Source data are provided as a Source data file.

To investigate whether the SAH process affects growth and expansion of the meningeal lymphatic vessels, we divided the meningeal lymphatics on the transverse sinus into eight different segments and measured the diameter and branching of each segment. Although we found significant dilation of vessel diameter (Fig. 7a, b) and a greater number of branches (Fig. 7a, c) in some segments in the SAH group, the lymphatic vessel area in the transverse sinus and the superior sagittal sinus did not change (Fig. 7d, e). Thus we conclude that lymphangiogenesis and lymphangiectasia of the meningeal lymphatics did not significantly alter at 7 days post-induction of SAH.

## Discussion

SAH most commonly occurs due to the rupture of an aneurysm in the cerebral artery, releasing blood into the SAS and resulting in a series of early neurological complications and DCI. Understanding the process by which meningeal lymphatics drain fluid, macromolecules and immune cells from the CSF has shed a light on the pathogenesis of neurological diseases, including Alzheimer's disease, multiple sclerosis, and ischemic brain injury in past 5 years[13,14,16,18,29,30]. However, whether the extravasated erythrocytes released into the CSF during SAH can be removed by meningeal lymphatics remains unclear. Here we show that the extravasated erythrocytes in the SAS are drained into dCLNs and mandibular LNs through the meningeal lymphatics and that the depletion of meningeal lymphatics blocks the clearance of extravasated blood.

During SAH, blood pours into the SAS, where erythrocytes break down and release hemoglobin (Hgb) and its products that contribute to brain injury[31,32]. Previous studies report that extravasated erythrocytes and their degradation products in the SAS can be cleared via mechanisms of clot lysis or phagocytosis[7,9,10,33]. With the upregulation of intercellular adhesion molecule-1 in cerebral blood vessel endothelial cells, macrophages and neutrophils enter the SAS and then phagocytose extravasated erythrocytes and Hgb[34–36]. It is proposed that macrophages and neutrophils are trapped in the SAS after phagocytosis of extravasated erythrocytes and Hgb, which then die and are degranulated within 2–4 days[33,35]. However, the structure and function of meningeal lymphatics have been defined recently[11,12]. Furthermore, microglia also play a role to clear in SAH by expressing heme oxygenase-1[37]. Microglial activation and monocyte infiltration are observed 24 and 72 h after SAH, respectively[38]. Here we show that erythrocytes can be detected in the CLNs (dCLNs and mandibular LNs) and meningeal lymphatics at 4 h post SAH, a time much early than macrophage/microglia-mediated clearance. These findings reveal that the extravasated erythrocytes can be drained into CLNs before being degraded into Hgb or phagocytosed by macrophages and neutrophils, at least in the very early stages of SAH (4 h after SAH in this study). Meanwhile, we found here that the depletion of meningeal lymphatics significantly blocked the drainage of extravasated erythrocytes into CLNs, demonstrating that meningeal lymphatics serve as the route of draining for erythrocytes into the CLNs. The blood clots were observed in the brain of SAH mice with ablated lymphatics indicating that the clot clearance may be affected by the reduced cerebral lymphatic drainage. As CSF flow is not unidirectional, ventricular blood presenting in some acute SAH is proposed that is refluxed from cisternal hemorrhage and not indicative of primary ventricle bleeding[39,40]. Thus we considered that the clots on the pons and medulla in this study may result from the ventricular blood refluxed from SAS. There are several potential routes of extravasated blood clearance as discussed above. Different routes may participate in different phases of SAH, for example, before and after the degradation of erythrocytes. Great effort should be made in the future to clarify the kinetics and relative contribution of each of these pathways of extravasated erythrocyte clearance.

This study reveals a lymphatic route for clearing extravasated erythrocytes in SAH; however, the mechanism by which extravasated erythrocytes enter the meningeal lymphatics is still unclear. T cells and dendritic cells enter into and migrate through the meningeal lymphatics and peripheral lymphatic system via the C-C chemokine motif receptor 7–C-C chemokine motif ligand 21 pathway[14,18,41,42]. Macromolecules can be endocytosed by brain lymphatic endothelial cells in zebrafish[43]. Fluids and solutes diffuse into the peripheral lymphatic lumen due to the pressure difference between that of the interstitial fluid and the lumen[44], while chylomicron uptake into lacteals occurs by active transport vesicles through lymphatic endothelial cells[45,46]. Little is known about the mechanism of extravasated erythrocytes entering meningeal lymphatics, but the known ways by which immune cells and macromolecules enter into the lymphatic lumen may provide some hints for future research.

Neuroinflammation that occurs as a result of SAH is caused by the accumulation of erythrocyte degradation products, including Hgb, methemoglobin, heme, and hemin in the SAS. Microglia are activated and accumulated in the brain accompanied by upregulation of inflammatory cytokines, including tumor necrosis factor-α, interleukin (IL)-1β, and IL-6. These inflammatory factors contribute to the neuronal cell death and secondary brain

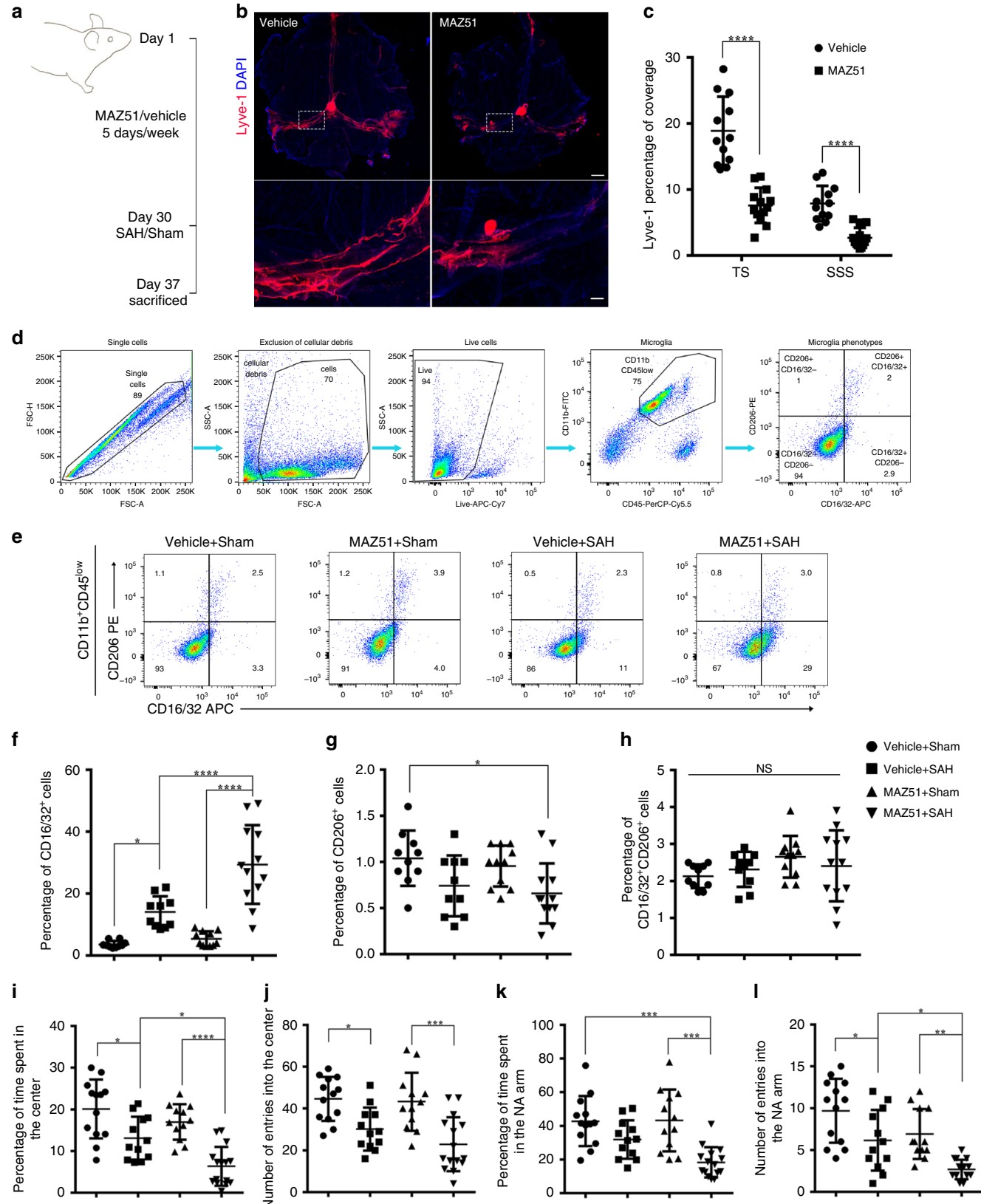

injury after SAH[38,47–49]. In this study, we found that microglial activation and polarization into a pro-inflammatory microglial cells in SAH are aggravated by the depletion of meningeal lymphatics. This exacerbation may have occurred because the

injury after SAH[38,47–49]. In this study, we found that microglial activation and polarization into a pro-inflammatory microglial cells in SAH are aggravated by the depletion of meningeal lymphatics. This exacerbation may have occurred because the extravasated erythrocytes were trapped in the SAS, thus increasing the accumulation of Hgb and its products and prolonging brain exposure to these degradation products. A previous study reported that the subarachnoid clot volume and spontaneous

**Fig. 5 The blockage of VEGFR3 exacerbated the neuroinflammation and behavioral defects induced by SAH. a** Scheme of the experiments in VEGFR3 tyrosine kinase inhibitor treatment experiments. Mice were treated with MAZ51 once a day for 30 days (5 days per week). After 7 days of SAH induction or sham treatment, mice were killed for analysis. **b** Representative images of meningeal lymphatics labeled by Lyve-1 (red) in the vehicle and MAZ51 treatment groups. Regions of interest in the top images are shown below. Scale bar, 1 mm (the top images) and 200 μm (the bottom images). **c** Quantification of the percentage of Lyve-1 coverage on TS and SSS. (Vehicle; $n = 12$, MAZ51; $n = 13$ mice. TS, $P < 0.0001$, SSS, $P < 0.0001$, two-tailed unpaired Student's $t$ test). **d** Representative FACS plots showing gating strategy in flow cytometric analysis. Cell populations characterized by $CD11b^+$ $CD45^{low}$ (microglia) were gated for further analysis. **e** Representative dot plots showing the ratio of CD16/32-positive subsets and CD206-positive subsets. **f** Quantification of $CD16/32^+CD206^-$ subsets of $CD11b^+CD45^{low}$ populations ($P$(Vehicle + Sham vs Vehicle + SAH) = 0.0139, $P$(MAZ51 + Sham vs MAZ51 + SAH) < 0.0001, $P$(Vehicle + SAH vs MAZ51 + SAH) < 0.0001), **g** $CD206^+CD16/32^-$ subsets of $CD11b^+CD45^{low}$ populations ($P$(Vehicle + Sham vs MAZ51 + SAH) = 0.0233), and **h** $CD16/32^+CD206^+$ subsets of $CD11b^+CD45^{low}$ populations in different groups. Vehicle + Sham/Vehicle + SAH; $n = 10$, MAZ51 + Sham; $n = 11$, MAZ51 + SAH; $n = 12$ mice, pooled from 2 independent experiments. **i–l** VEGFR3 blockage worsened neurological performance of SAH. **i** Percentage of time spent in the center area in the open field test ($P$(Vehicle + Sham vs Vehicle + SAH) = 0.011, $P$(MAZ51 + Sham vs MAZ51 + SAH) < 0.0001, $P$(Vehicle + SAH vs MAZ51 + SAH) = 0.0111), **j** the number of entries into the center area in the open field test ($P$(Vehicle + Sham vs Vehicle + SAH) = 0.0216, $P$(MAZ51 + Sham vs MAZ51 + SAH) = 0.0004), **k** percentage of time spent in the novel arm (NA) in the Y-maze test ($P$(Vehicle + Sham vs MAZ51 + SAH) = 0.0001, $P$(MAZ51 + Sham vs MAZ51 + SAH) = 0.0001), and **l** the number of entries into the NA in the Y-maze test in the groups of mice indicated ($P$(Vehicle + Sham vs Vehicle + SAH) = 0.0269, $P$(MAZ51 + Sham vs MAZ51 + SAH) = 0.0037, $P$(Vehicle + SAH vs MAZ51 + SAH) = 0.0221). Vehicle + Sham; $n = 13$, Vehicle + SAH/MAZ51 + Sham; $n = 12$, MAZ51 + SAH; $n = 15$ mice. All data are presented as mean values ± SD, one-way ANOVA with Turkey's multiple-comparison test (**f–l**), *$P < 0.05$, **$P < 0.01$, ***$P < 0.001$, ****$P < 0.0001$. NS, not significant. Source data are provided as a Source data file.

clearance rate are closely related to vasospasm[8]. Thus promoting the clearance of subarachnoid erythrocytes via accelerating meningeal lymphatic flow may be a potential therapy for the pathologies associated with SAH.

The enhancement of lymphatic draining function, lymphangiogenesis, and lymphangiectasia are commonly observed in peripheral acute inflammation, including arthritis, bacterial keratitis, and colitis[50–52], with the enhanced lymphatic flow and lymphangiogenesis reducing the local inflammation and edema. The overexpression of VEGF-C by viral infection or local injection has been shown to ameliorate, while the blockade of the VEGF-C/VEGFR-3 pathway exacerbates, inflammation[52–56]. Increasing meningeal lymphatic drainage was observed at day 7 of SAH in this study; however, lymphatic expansion and growth were not pronounced. These results are in consistent with previous reports that the morphology of meningeal lymphatics did not change in EAE-associated neuroinflammation[14,18]. Meningeal lymphatics may have a limited growth capability when exposed to neuroinflammation. Further research is needed to determine whether modulating lymphangiogenesis by overexpression of VEGF-C or other means affects SAH induced-neuroinflammation.

In summary, we show that extravasated erythrocytes in the SAS are drained into CLNs through meningeal lymphatics during SAH. This study adds insight into the extravasated erythrocyte clearance pathway that occurs during the very early stages of SAH and provides a possible therapeutic avenue for its treatment, as well as possibly other types of intracranial hemorrhage.

## Materials and methods

**Animals**. Specific pathogen-free, C57BL/6 male mice (6–8 weeks old) were purchased from Shanghai Model Organisms Center. Mice were housed in the animal facility with controlled habituation and temperature, on 12-h light vs dark cycles, and fed with regular rodent's chow and sterilized tap water ad libitum. Mice were allowed to accommodate for 2 weeks before experimental procedures. All animal procedures were approved by Longhua Hospital - Animal Ethics Committee and were performed according to the Guiding Principles for the Care and Use of Laboratory Animals Approved by Animal Regulations of National Science and Technology Committee of China.

**Induction of SAH**. An SAH model was established according to a previous publication[57]. Briefly, 60 μl autologous blood was withdrawn from the right femoral artery after mice were anesthetized with ketamine hydrochloride (100 mg/kg, Fujian Gutian Pharma Co., Ltd, 1505223). The animal's head was fixed in a stereotactic frame (RWD), the posterior neck was incised, the posterior neck muscles were separated to access the cisterna magna, and 60 μl of autologous blood was

injected at a low rate into this region. The needle was kept in place for 2 min to prevent backflow or CSF leakage. Sham-treated mice were similarly injected with 60 μl of saline. Then the mice were sutured and kept on a 37 °C heating pad (Thermo Plate) until entirely recovered from anesthetization. Mice in blank control group (Con group) were housed as usual and did not receive any additional treatment.

**Behavioral analysis**. The open field test was used to evaluate spontaneous activity and exploration behaviors. Mice moved freely in the box (60 cm × 60 cm × 25 cm) for 10 min, and then the distance traveled, the time spent in the center, and the number of entrances into the center area was recorded using the videotracking software EthoVision XT 12 (Noldus). Short-term memory was assessed by Y-maze test. The maze included the starting arm, the NA, and the other arm. Before the test, mice underwent a 5-min training period with a block of the NA in the maze. Two hours later, the NA was opened, and the mice were allowed to travel freely throughout the three arms, with the percentage of time spent in the NA and the number of entries into the NA in 5 min recorded by ANY-maze (Stoelting, America).

**I.c.m. injection**. Mice were fixed in a stereotactic frame (RWD) after anesthetization with ketamine hydrochloride, and an incision was done along with separation of the posterior neck muscles to access the cisterna magna. Then 2 μl of fluorescent microbeads (Latex beads, amine-modified polystyrene, fluorescent red, Cat. No. L2778-1, Sigma) were injected into the cisterna magna at a rate of 0.5 μl/min or 5 μl of Alexa Fluor 488-conjugated anti-Lyve-1 antibody (AF488 anti-Lyve-1) (Cat. No. 53-0443-82, eBioscience) was injected into the cisterna magna at a speed of 1 μl/min. After injection, the needle was left in place for 2 min to prevent backflow and leakage. Then the mice were sutured and kept on a 37 °C heating pad until responsive. The $AF^{488}$ Lyve-1 antibody was left to flow for 30 min and the fluorescent microbeads were left to flow for 2 h before the mice were killed.

**Visudyne treatment**. To ablate the meningeal lymphatics, visudyne treatment was carried out according to a previous publication[14]. Briefly, mice were anesthetized with ketamine hydrochloride and their heads were fixed in a stereotactic instrument. Visudyne (APExBIO, Cat. No. A8327) was reconstituted according to the manufacturer's instructions, and 5 μl was injected into the cisterna magna at a speed of 1 μl/min. Fifteen minutes later, a nonthermal 689-nm wavelength laser light (Changchun Laser Technology), with a dose of 50 J/cm² and intensity of 600 mW/cm², was applied on 5 different spots through the skull (the injection site, left and right transverse sinuses, the superior sagittal sinus, and the junction of all sinuses). For the Laser group, mice underwent the same procedures of laser treatment but without visudyne injection, and for the Visudyne group, mice were just given 5 μl of visudyne into the cisterna magna without laser treatment. During the laser procedure, the eyes of the mice were protected. Then the incision in the mice was sutured and the mice were kept on a 37 °C heating pad until entirely recovered from anesthetization.

**VEGFR3 tyrosine kinase inhibitor administration**. MAZ51 (Cat. No. 676492, Merck Millipore) was dissolved in dimethyl sulfoxide and intraperitoneally injected at 10 mg/kg of body weight for 30 days (5 days per week). The control group was

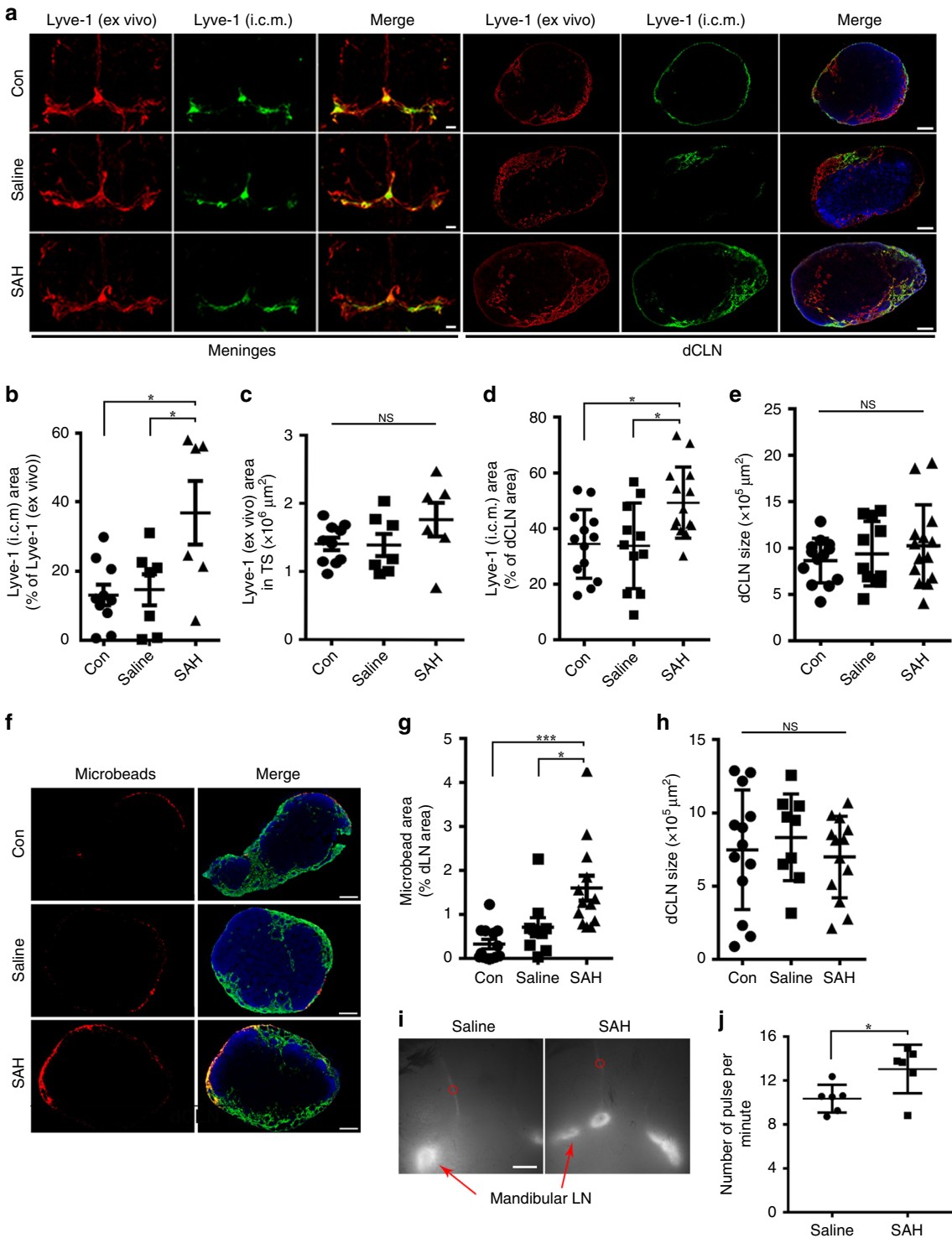

given the same volume of vehicle. On the 30th day, mice of both groups were divided into either an autologous blood injection or a saline injection group. On the seventh day after SAH induction, mice were killed for analysis.

**Erythrocyte isolation and labeling**. Whole blood was collected from mice from the right femoral artery after they were anesthetized, then 1:1 diluted with 2% fetal bovine serum (FBS)–phosphate-buffered saline (PBS), followed by centrifugation for 10 min ($800 \times g$) without braking. The plasma and buffy coat layers were removed, the erythrocytes were collected from the bottom of tubes, and the cells were diluted into $10^6$ cells/ml, then CFSE (20 µM/ml, eBioscience, Cat. No.

65-0850-84) was added, followed by incubation for 10 min in 37 °C. After washing with 2% FBS–PBS, erythrocytes (about $10^6$ cells in 60 µl) were injected into the cisterna magma. The control group mice were injected with saline. Four hours after erythrocyte was injected, meninges, dCLNs, and mandibular LNs were harvested.

**Flow cytometry**. Mice brains were dissected after transcardial perfusion by cold PBS, then minced into small pieces. Brain tissue was digested by collagenase A (1 mg/ml, Sigma Aldrich, Cat. No. 10103578001) for 30 min at 37 °C, then filtered by 70-µm nylon mesh cell strainers (BD bioscience). A cell suspension was made with 30% stock isotonic percoll (SIP) (GE, 17089109) and layered on the top of

**Fig. 6 Meningeal lymphatic flow is augmented in the SAH mouse model. a** Representative images of meninges and dCLNs stained for anti-Lyve-1 (AF[555]-conjugated secondary antibody, ex vivo) and AF[488]-conjugated anti-Lyve-1 (i.c.m.). Scale bars, 1 mm (for the meninges,); 200 μm (for the dCLNs). $n = 2$ independent experiments. **b** Quantification of area fraction (%) dividing the area of meningeal lymphatics labeled by AF[488]-conjugated Lyve-1 antibody (i.c.m.) by the area of meningeal lymphatics (Con; $n = 10$, Saline; $n = 7$, SAH; $n = 6$ mice. $P$(Con vs SAH) = 0.0122, $P$(Saline vs SAH) = 0.0303). **c** Quantification of meningeal lymphatic area of the TS. Con; $n = 10$, Saline; $n = 7$, SAH; $n = 6$ mice. **d** Quantification of area fraction (%) occupied by the area of dCLN lymphatics labeled by AF[488]-conjugated Lyve-1 antibody (i.c.m.) and the area of dCLNs. Con; $n = 13$, Saline; $n = 11$, SAH; $n = 14$ nodes, pooled from 2 independent experiments. $P$(Con vs SAH) = 0.0183, $P$(Saline vs SAH) = 0.0181). **e** Size of dCLNs in the Con, saline, and SAH groups. Con; $n = 13$, Saline; $n = 11$, SAH; $n = 14$ nodes, pooled from 2 independent experiments. **f** Representative images of exogenously injected fluorescent microbeads (1 μm in diameter, red) in the dCLNs of the control, saline injection, and autologous blood injection groups. Scale bar, 200 μm. $n = 2$ independent experiments. **g** The percentage of microbead coverage in the dCLNs. Con/SAH; $n = 13$, Saline; $n = 9$ nodes, pooled from 2 independent experiments. $P$(Con vs SAH) = 0.0004, $P$(Saline vs SAH) = 0.0251). **h** Size of dCLNs in the Con, saline, and SAH groups. Con/SAH; $n = 13$, Saline; $n = 9$ nodes, pooled from 2 independent experiments. **i** Representative images of ICG fluorescence of mandibular LNs and its afferent lymphatics at 30 min after ICG injection. Scale bar, 3 mm. Red circle, region of interest of lymphatic vessel for lymph flow frequency analysis. **j** Quantification of the number of the afferent lymphatic vessel contraction frequency (pulse per minute) ($n = 6$ mice per group, $P = 0.0261$, two-tailed unpaired Student's $t$ test). All data are presented as mean values ± SD; one-way ANOVA with Turkey's multiple-comparison test (**b–e**, **g**, **h**), *$P < 0.05$, ***$P < 0.001$. NS, not significant. Source data are provided as a Source data file.

70% SIP and then centrifuged at $500 \times g$ at 25 °C for 30 min without braking. Cells were collected from the 70–30% SIP interphase and stained for live cells by Fixable Viability Dye eFluor™ 780 (Cat. No. 65-0865-18, eBioscience), extracellular markers with the following antibodies at a 1:100 dilution: rat anti-CD11b fluorescein isothiocyanate (FITC)-conjugated antibody (11-0112-82, eBioscience), rat anti-CD45 PerCP-Cy5.5-conjugated antibody (45-0451-82, eBioscience), rat anti-CD16/32 allophycocyanin (APC)-conjugated antibody (558636, BD Bioscience) and intracellular marker rat anti-CD206 R-phycoerythrin (PE)-conjugated antibody (12-2061-80, eBioscience). The corresponding isotype control antibodies that were used are as follows: Rat IgG2b κ Isotype control FITC-conjugated antibody (11-4031-82, eBioscience) Rat IgG2a κ Isotype control PerCP-Cy5.5-conjugated antibody (45-4321-80, eBioscience), Rat IgG2b κ Isotype control PE-conjugated antibody (12-4031-82, eBioscience), and Rat IgG2b κ Isotype control APC-conjugated antibody (553991, BD Bioscience). Samples were tested and analyzed by Longzoe (Shanghai) Biotechnology Co., Ltd using BD Fortessa X20 and the FlowJo V10 software, and the company was blinded to the group allocations.

**Laser speckle**. Mice were anesthetized by isoflurane, an incision was done along the midline to separate the skin of the skull, and RFLSI Pro+ laser speckle (RWD Life Science Co., Ltd) was used to detect mice cerebral blood flow. Laser speckle blood flow images were recorded and used to identify the regions of interest (ROIs). Within these ROIs, the mean blood flow index was calculated in real time.

**In vivo imaging**. Mice were fixed in a stereotactic frame (RWD) after anesthetization with ketamine hydrochloride, an incision was performed, and the posterior neck muscles were separated to access the cisterna magna. Five μl of visudyne was injected into the cisterna magna at a speed of 1 μl/min, and the needle was kept in place for 2 min to avoid leakage. Control group mice were not injected with any solution. Fifteen minutes later, the distribution of visudyne was detected by KODAK In-Vivo Multispectral Imaging System FX using a 630-nm laser for excitation. Then mice were killed to acquire the skulls, and the visudyne distributions on the skull were also recorded.

**Indocyanine green near-infrared (ICG-NIR) imaging**. ICG was dissolved in saline (2 mg/ml, Cat. No. 17478-701-02, Akorn). Mice from the SAH group (at 7 days post-surgery) and the saline-injected group were fixed in a stereotactic frame (RWD) after anesthetization, and cisterna magna was exposed. Five μl of ICG was injected into the cisterna magna (1 μl/min), and then the needle was left in place for 2 min to avoid leakage. ICG fluorescence of mandibular LNs and its afferent lymphatics were visualized by an IR laser (Changchun Laser Technology) and recorded continuously by an Olympus microscope (exposure times 200 ms) for 1 h. The images were analyzed using the Image J software. ROIs were identified in the afferent lymphatic vessel, and vessel contraction rate (pulse/min) was calculated to present the lymph flow function according to previous studies[20,58].

**Tissue processing**. dCLNs and mandibular LNs were harvested in a deep anesthesia condition, fixed in 4% paraformaldehyde (PFA) overnight, and then incubated serially in 10%, 20%, and 30% sucrose solutions for 3 days each. For immunofluorescence staining, CLNs were embedded in OCT, and 7-μm-thick sections were sliced by a cryostat (Leica, CM3050S). After transcardial perfusion with saline and 4% PFA for 15 min, the skullcap was harvested and fixed in 4% PFA overnight, and then the meninges were dissected from the skullcap.

**Immunofluorescence**. For immunofluorescence, the whole mounts and sections were blocked by 0.3% PBST with 5% bovine serum albumin for 1 h at room temperature, then incubated with primary antibodies overnight at 4 °C. After washing with PBS three times for 15 min each, secondary antibodies were incubated for 2 h at room temperature. Finally, the whole mounts and sections were mounted with mounting medium with 4,6-diamidino-2-phenylindole (Cat. No. F6057, Sigma). The primary antibodies used in immunofluorescence included rabbit anti-Lyve-1 antibody (1:1000; Abcam, Cat. No. ab14917), rat anti-Ly76 [Ter119] antibody (1:500; Abcam, Cat. No. ab91113), rat anti-Ter119 PE-conjugated antibody (1:100; Cat. No. 12-5921-81, eBioscience), rat anti-Lyve-1 eFluor 660-conjugated antibody (1:200; 50-0443-80, eBioscience), hamster anti-PDPN antibody (1:200; Abcam, Cat. No. ab11936), rabbit anti-Prox1 antibody (1:100; AngioBio, Cat. No. 11-002P), and rat ant-CD31 antibody (Abcam, Cat. No. ab7388). The corresponding secondary antibodies were used as follows: Dylight™ 488-labeled goat anti-rabbit antibody (1:200; KPL, Cat. No. 072-03-15-06), Dylight™ 488-labeled goat anti-rat antibody (1:200; Cat. No. 072-03-16-06, KPL), Alexa Fluor 546 goat anti-hamster antibody (1:200, Invitrogen, Cat. No. A-21111), Alexa Fluor 488/555 goat anti-rat antibody (1:1000, Cell Signaling Technology, Cat. No. 4416S/4417S), and Alexa Fluor 555 goat anti-rabbit antibody (1:1000, Cell Signaling Technology, Cat. No. 4413S).

**Image analysis**. For the whole-mount staining of meninges, images were acquired with an Olympus VS120 microscope and a 10× objective with 0.4 NA, or acquired with an Olympus FV1000 confocal microscope and a 40× objective with 0.95 NA, with a resolution of 1024 × 1024 pixels and a z-step of 4 μm. The exposure time and brightness/contrast of each image were applied equally across all images, and images were analyzed using the Image J (NIH) software. For the CLN sections, images were acquired with an Olympus VS120 microscope and a 20× objective with 0.75 NA.

The numbers of erythrocyte per mm$^2$ in the Lyve-1-positive lymphatic sinus of CLNs were calculated, with only the erythrocytes with intact morphology included. The mean value of five sections of each CLN was used to make a plot graph. The numbers of erythrocytes in meningeal lymphatics per field were calculated, and four to five fields of each meninges were quantified to acquire the mean value. The percentage of meningeal lymphatics labeled by AF488 Lyve-1 antibody (i.c.m.) was defined by dividing the area of AF488 Lyve-1 antibody (i.c.m.) labeled by the area of meningeal lymphatics. The percentage of dCLN lymphatics labeled by AF488 Lyve-1 antibody (i.c.m.) was determined by dividing the area AF488 Lyve-1 antibody (i.c.m.) labeled per section by the area of the dCLN section. The mircobead coverage in dCLN was quantified by dividing the area of microbeads per section by the area of the dCLN section. Five to ten sections of each dCLN were quantified to acquire the mean value. Lymphatic ablation and lymphatic regression were measured by dividing the area of Lyve-1 labeled by the area of the sinus, and lymphatic coverage on transverse sinus and sagittal sinus was calculated separately. Percentage of blood vasculature coverage on sinuses was calculated by dividing the area of the CD31-positive vessels by the area of sinuses. Raw data were collected using the Microsoft Excel 2007 software.

**Statistical analysis**. Data were expressed as means ± SD, with differences between mean values determined by two-tailed unpaired Student's $t$ test, one-way analysis of variance (ANOVA), or two-way ANOVA with Turkey's multiple-comparison test using the GraphPad Prism 6 Software. $P$ values < 0.05 were considered significant. The investigators responsible for data analysis were blinded to the group allocations.

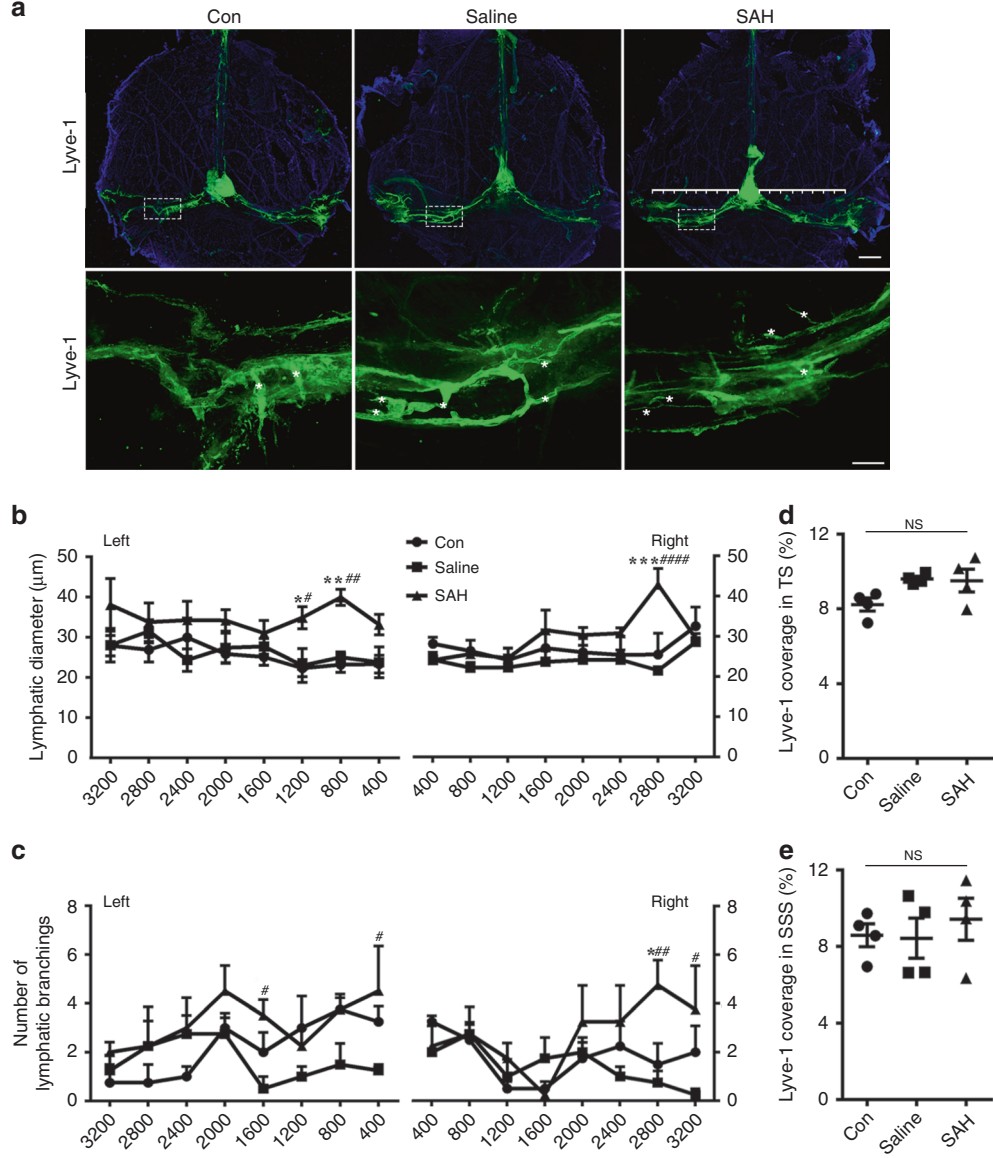

**Fig. 7 The diameter, branching, and area of meningeal lymphatics do not change significantly after 7 days of SAH. a** Representative images of meninges whole mounts stained for meningeal lymphatics (Lyve-1, green) from the indicated groups (4 mice per group). The lymphatics on left and right TS were divided into 8 segments (400 μm each), respectively. Regions of interest of each top image are shown in the row below (white asterisk, the branching of lymphatics). Scale bars, 1 mm (the top images) and 200 μm (the bottom images). **b** Quantification of lymphatic diameters in eight different segments. X axis represents 400–3200 μm distance to the junction of all sinuses. $n = 4$ mice per group. Left 800, $P$(Con vs SAH) = 0.002, $P$(Saline vs SAH) = 0.0067; Left 1200, $P$(Con vs SAH) = 0.0262, $P$(Saline vs SAH) = 0.0374; Right 2800, $P$(Con vs SAH) = 0.0005, $P$(Saline vs SAH) < 0.0001. **c** Quantification of lymphatic branches in eight different segments. X axis represents 400–3200 μm distance to the junction of all sinuses. $n = 4$ mice per group. Left 1600, $P$(Saline vs SAH) = 0.0473; Left 400, $P$(Saline vs SAH) = 0.0287; Right 2800, $P$(Con vs SAH) = 0.0319, $P$(Saline vs SAH) = 0.0063; Right 3200, $P$(Saline vs SAH) = 0.0191. **d, e** The percentages of Lyve-1 coverage on the TS (**d**) and the SSS (**e**). $n = 4$ mice per group. All data are presented as mean values ± SD; two-way ANOVA (**b, c**) or one-way ANOVA (**d, e**) with Turkey's multiple-comparison test. *$P < 0.05$, **$P < 0.01$, ***$P < 0.001$, Con vs SAH, #$P < 0.05$, ##$P < 0.01$, ####$P < 0.0001$, Saline vs SAH. NS, not significant. Source data are provided as a Source data file.

**Reporting summary**. Further information on research design is available in the Nature Research Reporting Summary linked to this article.

## Data availability

The raw data underlying Figs. 1c, e, 2b, e, 3b, c, f, 4c–i, 5c, f–l, 6b–e, g, h, j, 7b–e and Supplementary Figs. 1d and 3b, d, h, j are available via a source data file submitted with this manuscript. All other data are available from the corresponding authors upon reasonable request.

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

## Acknowledgements

We express gratitude to Fudan University for access to their confocal microscope and their mice behavioral analysis platforms. And thanks to Dr. Weian Zhang for the in vivo

imaging and Professor Beihua Zhang for the laser speckle imaging. We are also grateful to Professor Baohua Tian for his help in the intact erythrocytes and the degradant differentiation. This work was sponsored by research grants from National Key R&D Program of China (2018YFC1704300 to Y.W.), National Natural Science Foundation (81822050 and 81920108032 to Q.L., 81904227 to Y.W.), Leading medical talents in Shanghai (2019LJ02 to Q.L.), Dawn plan of Shanghai Municipal Education Commission (19SG39 to Q.L.), the program for innovative research team of Ministry of Science and Technology of China (2015RA4002 to Y.W.), "Innovation Team" development projects (IRT1270 to Y.W.), Shanghai TCM Medical Center of Chronic Disease (2017ZZ01010 to Y.W.), Three Years Action to Accelerate the Development of Traditional Chinese Medicine Plan (ZY(2018-2020)-CCCX-3003 to Y.W.), and the program of Longhua Hospital (KY1932 to Y.W.).

## Author contributions

J.C., L.W., H.X., L.X., Q.L., and Y.W. conceived and designed the study; J.C., L.W., S.C., and Z.G. performed the experiments; Y.Z. performed behavioral tests and data analysis; X.L. analyzed the data of blood flow index and counted the numbers of erythrocytes in CLNs and in meningeal lymphatics; Z.Z. analyzed the data of lymph flow frequency, counted the numbers of erythrocytes in CLNs, and calculated the percentage of meningeal lymphatics and blood vasculature coverage on sinus; C.W. counted the numbers of erythrocyte in CLNs and analyzed the area of AF[488] Lyve-1 antibody (i.c.m.) labeled lymphatics and mircobead coverage (Y.Z., X.L., Z.Z., and C.W. were blinded to group allocations); J.C. and Q.L. drafted the manuscript; L.W., H.X., L.X., Q.L., and Y.W. revised the manuscript. All authors have approved the final version of the manuscript and have agreed to be accountable for all aspects of the work.

## Competing interests

The authors declare no competing interests.
