## [Peer Review File · Nature Communications]

Reviewers' Comments:

Reviewer #1:

Remarks to the Author:

The manuscript by Cehn et al., entitled "Meningeal lymphatics clear erythrocytes caused subarachnoid hemorrhage", presents a new evidence for the role of meningeal lymphatic vessels in subarachnoid hemorrhage. The role of meningeal lymphatic vessels has been recently demonstrated in mouse models of AD, MS mouse model of EAE, as well as in models of PD, etc. Here the authors provide a compelling evidence that meningeal lymphatic vessels are impacted by the subarachnoid hemorrhage and that impairment of their function exacerbates the outcome. The experiments are well-performed. It would have been interesting to see whether VEGF-c could improve the outcome of subarachnoid hemorrhage but these results may indeed be outside of the scope of the current work. I do not have any requests for additional experiments, except for repeat of the existing ones to solidify their data. The authors do not mention how many times each experiment was reproduced (or at least I missed this information) and the number of mice used in each group is low. The authors should repeat their studies to have at least 12 animals per groups combined. The authors also do not mention whether the experiments were blinded or not. It is extremely important for the authors to repeat their experiments with higher number of mice and to make sure the groups are blinded.

Reviewer #2:

Remarks to the Author:

Chen et al present the results of experiments aimed to determine the role of the meningeal lymphatic system on clearance of erythrocytes after experimental subarachnoid hemorrhage in mice. Resolution of subarachnoid hemorrhage is an important mechanism preventing secondary brain injury. The role of the lymphatics in this process would have therapeutic implications and add to the growing body of literature on the importance of the meningeal lymphatic system in neurological disease. However, multiple concerns limit enthusiasm for the paper. Overall, more experimental rigor is needed throughout in order to convince readers of their conclusions.

1. Figure 1A- Why are deep cervical lymph nodes smaller after SAH and NaCl injection than in control? The figure legend states the scale bar is 500 μ m but this is impossible. These representative images also do not match the immunofluorescence in 1B or quantification in 1D.

2. Figure 1 E is the only figure to attempt to show erythrocytes in the meningeal lymphatics. Since their primary claim is that the meningeal lymphatics are being used to clear the erythrocytes, showing erythrocytes in these structures is critical. However there is no control (non-SAH) image, only a single image is shown, there is no quantification, and the Lyve1 cells are often co-localized with erythrocytes suggesting engulfment (which raises the question whether they are actually staining some macrophages, which are also positive for Lyve1).

3. They use visudyne injected into the cisterna magna followed by photoablation to determine the effect of lymphatic ablation on erythrocyte clearance, inflammation, and behavior. These experiments depend on the selective ablation of lymphatics without changes to venous sinus flow and damage to arachnoid granulations and therefore the direct connection from CSF to the venous system. What is the distribution of visudyne at 15 minutes after injection? Confirmation that this approach has no effect on the other major structures in the region is critical.

4. What does visudyne + photoactivation do to inflammation and mouse behavior in the absence of SAH? All experiments need this control in order to interpret the results.

5. Since all the main results and conclusions of the paper rely on their interpretation of the photoablation experiments, an additional method of manipulating lymphatics should be performed

to confirm the results are not due to off target effects. Several methods for manipulation of the VEGFC system have been published to complement their current approach.

6. Why were different time points for anti-Lyve1 antibody and fluorescent microbeads used in Figure 5?

7. Are erythrocytes found only in the deep cervical lymph nodes that drain CSF? Did they examine other lymph nodes to confirm the specificity of their localization?

8. Were behavioral outcomes analyzed blinded to treatment group?

Reviewer #3:

Remarks to the Author:

This is a very interesting article that attempts to address how red blood cells may be cleared by cerebral lymphatics. The authors do a significant amount of immunohistochemistry to try to prove their point, but in the end there are many critical flaws with this manuscript.

Major

1. Using Ter119, while it is specific to murine red blood cells, does not mean that the rbc's are still intact when they are seen in the dCLNs. Ter119 may be recognizing a degraded rbc membrane.

2. In fig 2E, to attempt to prove that the meningeal lymphatics are draining the rbc's, a laser visudyne system is used, but the control dCLNs look much different compared to an SAH model alone. The lyve and ter119 look to be partitioned across the dCLNs. The authors make no mention of this.

3. Using the area of Ter119 is not quantitative, it is qualitative at best. What if the reason one is seeing more ter119 is because the rbc's are being broken down more effectively and the plasma membranes are spreading out across the dCLNs? The rbc's should be quantified stereologically.

4. In figure 3, the increased neuroinflammation should be quantified by flow cytometry, not immunohistochem and western blot. Many myeloid cells can express Iba1 and it does not mean that they are reactive.

5. In figure 4, the authors give the impression that ablation of lymphatics leads to worsened cognitive performance; however, no post hoc tests show a difference between L+V+SAH and V+SAH.

6. I don't understand how dynamic lymphatic flow can be quantified based a static picture. This needs more explanation, ir it must be done by 2 photon microscopy.

Minor

There are numerous grammatical errors throughout the manuscript in tense and plurality.

All data should be shown as mean and standard deviation, not standard error of the mean, which is much smaller than the standard deviation.

Reviewers' comments:

Reviewer #1 (Remarks to the Author):

The manuscript by Cehn et al., entitled “Meningeal lymphatics clear erythrocytes caused subarachnoid hemorrhage”, presents a new evidence for the role of meningeal lymphatic vessels in subarachnoid hemorrhage. The role of meningeal lymphatic vessels has been recently demonstrated in mouse models of AD, MS mouse model of EAE, as well as in models of PD, etc. Here the authors provide a compelling evidence that meningeal lymphatic vessels are impacted by the subarachnoid hemorrhage and that impairment of their function exacerbates the outcome. The experiments are well-performed. It would have been interesting to see whether VEGF-c could improve the outcome of subarachnoid hemorrhage but these results may indeed be outside of the scope of the current work. I do not have any requests for additional experiments, except for repeat of the existing ones to solidify their data. The authors do not mention how many times each experiment was reproduced (or at least I missed this information) and the number of mice used in each group is low. The authors should repeat their studies to have at least 12 animals per groups combined. The authors also do not mention whether the experiments were blinded or not. It is extremely important for the authors to repeat their experiments with higher number of mice and to make sure the groups are blinded.

Response: thank you so much for the important advice.

We have repeated the main results and increased the number of mice in each group. The number of replicates for each experiment was added to the figure legends. The experiments were performed blinded and their descriptions have been added to the Methods section.

Reviewer #2 (Remarks to the Author):

Chen et al present the results of experiments aimed to determine the role of the meningeal lymphatic system on clearance of erythrocytes after experimental subarachnoid hemorrhage in mice. Resolution of subarachnoid hemorrhage is an important mechanism preventing secondary brain injury. The role of the lymphatics in this process would have therapeutic implications and add to the growing body of literature on the importance of the meningeal lymphatic system in neurological disease. However, multiple concerns limit enthusiasm for the paper. Overall, more experimental rigor is needed throughout in order to convince readers of their conclusions.

1. Figure 1A- Why are deep cervical lymph nodes smaller after SAH and NaCl

injection than in control? The figure legend states the scale bar is 500 μ m but this is impossible. These representative images also do not match the immunofluorescence in 1B or quantification in 1D.

Response: The sizes of deep cervical lymph nodes differ greatly in individual mice, thus looking at four to five nodes may be misleading. We therefore increased the sample size to 11 lymph nodes per group in the revised manuscript (Fig. 1A). We are sorry for a mistaken scale bar, and we have corrected it. And more representative images have replaced the former ones (Fig. 1B-C).

2. Figure 1 E is the only figure to attempt to show erythrocytes in the meningeal lymphatics. Since their primary claim is that the meningeal lymphatics are being used to clear the erythrocytes, showing erythrocytes in these structures is critical. However there is no control (non-SAH) image, only a single image is shown, there is no quantification, and the Lyve1 cells are often co-localized with erythrocytes suggesting engulfment (which raises the question whether they are actually staining some macrophages, which are also positive for Lyve1).

Response:

We have repeated this result, and control groups were included. In particular, meninges from a control group (no injection), a saline injection group and an autologous blood injection group (SAH) were included. The quantification of the number of erythrocytes in lymphatics per field was performed (shown in Fig. 1D-E). In the SAH group, the number of erythrocytes accumulated in the lymphatics was significantly higher than that in the other groups, while the saline injection group also showed a slight increase, which may be because the blood caused by the surgery infused into the cisterna magna from the saline injection site. The corresponding orthogonal views taken from a saline group or SAH group are shown in S. 2A-B. The orthogonal views show that the Ter 119⁺ erythrocytes of interest in the XY, XZ, and YZ planes co-localize with lyve-1⁺ lymphatic endothelial cells, suggesting these cells are drained by meningeal lymphatic vessels.

As noted, some macrophages are also positive for lyve-1, and thus we used other classical lymphatic endothelial cell markers to identify the lymphatics including Podoplanin and Prox1. As shown in Fig. 1F, Podoplanin and Prox1 are also co-localized with lyve-1⁺ and Ter119⁺ cells.

To further confirm the erythrocytes in meningeal lymphatics and CLNs were from exogenous injection, we labeled the erythrocytes ex vivo with CFSE and then injected them into the cisterna magna. Four hours post injection, meningeal lymphatics, dCLNs and mandibular LNs were also observed to show an accumulation of CFSE-labeled erythrocytes (Fig. 2), while the control group (saline injection) did not show any evidence of labeled erythrocytes in the meningeal lymphatics and LNs. The corresponding orthogonal views of CFSE-labeled (ex vivo) erythrocytes within lymphatic vessels are shown in S. 2C.

3. They use visudyne injected into the cisterna magna followed by photoablation to determine the effect of lymphatic ablation on erythrocyte clearance, inflammation,

and behavior. These experiments depend on the selective ablation of lymphatics without changes to venous sinus flow and damage to arachnoid granulations and therefore the direct connection from CSF to the venous system. What is the distribution of visudyne at 15 minutes after injection? Confirmation that this approach has no effect on the other major structures in the region is critical.

Response:

The distribution of visudyne at 15 minutes after injection was detected by *in vivo* fluorescence imaging. Visudyne is a kind of porphyrin, which can be excited by 630 nm laser light. The distribution the visudyne at 15 minutes *in vivo* is as follows (CON = no injection of any solution):

The distribution of the visudyne on the skull is shown in S. 3E.

The cerebral blood flow of mice was evaluated by the laser speckle. When comparing to normal mice, the cerebral blood flow of the mice that underwent visudyne and photoconversion was not altered (data shown in S. 3A-B). And the blood vasculature (labeled by CD31) coverage on sinuses in the visudyne+laser group was not changed when compared to laser only group and visudyne only group (data shown in S. 3C-D).

The VEGFR3 tyrosine kinase inhibitor experiments were carried out to further confirm our findings and avoid any off-target effects. MAZ51 is a VEGFR3 tyrosine kinase inhibitor, has been demonstrated to effectively inhibit lymphangiogenesis and has a high specificity for VEGFR3 with only partial inhibition of VEGFR1 and VEGFR2 at higher concentrations. The administration of MAZ51 also did not alter the cerebral blood flow (S. 3A-B) nor the blood vasculature coverage on sinuses (S. 3I-J).

4. What does visudyne + photoactivation do to inflammation and mouse behavior in the absence of SAH? All experiments need this control in order to interpret the results.

Response: We have repeated the experiments and included a visudyne + laser group to compare the differences among groups. We found that the visudyne + laser resulted in microglia cells activation and mouse behavioral defects, but the activated microglia

dramatically polarized to proinflammatory M1 phenotype and the behavioral defects aggravated after SAH (visudyne + laser vs visudyne + laser + SAH) (data shown in Fig. 4). The inflammation and behavioral defects caused by visudyne + photoactivation may due to the surgery, the injection of visudyne which is foreign to the mouse, the phototoxic of visudyne and so on. Microglia may sense even small imbalances of environmental homeostasis and are rapidly activated, and the activated microglia appear to be the predominant source for the inflammatory mediators in the central nervous system^{1,2}.

1. Ma Y, Wang J, Wang Y, Yang GY. The biphasic function of microglia in ischemic stroke. *Prog Neurobiol* 157, 247–72, (2017).

2. Aguzzi A, Barres BA, Bennett ML. Microglia: scapegoat, saboteur, or something else? *Science* 339, 156–61, (2013).

5. Since all the main results and conclusions of the paper rely on their interpretation of the photoablation experiments, an additional method of manipulating lymphatics should be performed to confirm the results are not due to off target effects. Several methods for manipulation of the VEGFC system have been published to complement their current approach.

Response: thank you so much for the important comment. We have performed the VEGFR3 tyrosine kinase inhibitor experiments to further confirm the notion that the depletion of meningeal lymphatics resulted in the aggravation of microglia activation and behavioral defects in SAH. MAZ51 is a highly selective VEGFR3 tyrosine kinase inhibitor and has been demonstrated effectively in inhibiting lymphangiogenesis^{1,2,3}. We found that after the administration of MAZ51 for 30 days, meningeal lymphatics regressed comparing to vehicle treatment group as shown in Fig. 5A-C. The microglia activation and behavioral defects were exacerbated in MAZ51 + SAH group (data shown in Fig. 5D-K). And these outcomes in MAZ51 + Sham group did not change significantly in comparison to the vehicle + Sham group.

1. Harding, J. et al. Lymphangiogenesis is induced by mycobacterial granulomas via vascular endothelial growth factor receptor-3 and supports systemic T-cell responses against mycobacterial antigen. *Am. J. Pathol.* 185, 432-445 (2015).

2. Kirkin, V. et al. Characterization of indolinones which preferentially inhibit VEGF-C- and VEGF-D-induced activation of VEGFR-3 rather than VEGFR-2. *Eur. J. Biochem.* 268, 5530-5540 (2001).

3. Anneli Ny, Marta Koch, Wouter Vandeveld, et al. Role of VEGF-D and VEGFR-3 in developmental lymphangiogenesis, a chemicogenetic study in *Xenopus* tadpoles. *Blood* 112,1740-1749 (2006).

6. Why were different time points for anti-Lyve1 antibody and fluorescent microbeads used in Figure 5?

Response: According to a previous study¹, the area of meningeal lymphatics labeled by anti-Lyve1 antibody increased quickly in 15 minutes post injection and reached a plateau at 30 minutes, so we chose the 15 minute after injection as the time point,

which may be more sensitive to differences among groups than the longer time point. But for the fluorescent microbeads, its diameter is 1 μm , which is drained much more slowly, and they were observed to be drained into CLNs at 2 hours after injection in a previous study¹.

1. Louveau, A., Herz, J., Alme, M. N., Salvador, A. F., et al. CNS lymphatic drainage and neuroinflammation are regulated by meningeal lymphatic vasculature. *Nat Neurosci* 21, 1380-1391, doi:10.1038/s41593-018-0227-9 (2018).

7. Are erythrocytes found only in the deep cervical lymph nodes that drain CSF? Did they examine other lymph nodes to confirm the specificity of their localization?

Response: We have examined other main lymph nodes located in the neck, forelimb and chest including mandibular LN, superficial parotid LN, axillary LN, brachial LN and tracheobronchial LN (data shown in S. 1A). We found that besides dCLN, the mandibular LN also drained erythrocytes in CSF, but the superficial parotid LN though located on the superficial anterior neck, did not drain the extravasated blood in CSF.

8. Were behavioral outcomes analyzed blinded to treatment group?

Response: The behavioral outcomes were analyzed by YKZ was blinded to group allocations.

Reviewer #3 (Remarks to the Author):

This is a very interesting article that attempts to address how red blood cells may be cleared by cerebral lymphatics. The authors do a significant amount of immunohistochemistry to try to prove their point, but in the end there are many critical flaws with this manuscript.

Major

1. Using Ter119, while it is specific to murine red blood cells, does not mean that the rbc's are still intact when they are seen in the dCLNs. Ter119 may be recognizing a degraded rbc membrane.

Response: Ter 119 do recognize the intact red blood cells and degraded rbc membrane, so after consultation with an expert we realized we could not discriminate the intact erythrocytes and their degraded membrane by staining different cell markers. With the higher resolution of images, we observed that there were clear morphologically intact erythrocytes and clusters of degraded membranes in the lymphatics of CLNs and meninges at the same time (data shown in Fig. 1B, D, white arrows, intact erythrocytes, blue arrows, degraded membrane), but we admit it is not clear yet that the erythrocytes were broken down before transportation by lymphatics or afterwards. But the data suggests that at least some intact erythrocytes were drained into CLNs before degradation.

2. In fig 2E, to attempt to prove that the meningeal lymphatics are draining the rbcs, a laser visudyne system is used, but the control dCLNs look much different compared to an SAH model alone. The lyve and ter119 look to be partitioned across the dCLNs. The authors make no mention of this.

Response: We have repeated these results and used more representative images to replace the old ones (data shown in Fig. 3E). We considered that the image of laser + visudyne group looked different from that of laser group and visudyne group in the old Fig 2E, which may be because the selective images were not from similar sections of dCLNs.

3. Using the area of Ter119 is not quantitative, it is qualitative at best. What if the reason one is seeing more ter119 is because the rbcs are being broken down more effectively and the plasma membranes are spreading out across the dCLNs? The rbcs should be quantified stereologically.

Response: To address this problem, we counted the number of erythrocytes with clear morphology in dCLNs and mandibular LNs. Five sections of each CLN were counted, the mean value of the number of erythrocytes per mm^2 of all sections from the same CLN was calculated and used for statistical analysis. We hope that this method would be more quantitative (data shown in Fig. 1B-C, Fig. 3E-F, S. 1C-D, S. 3G-H).

4. In figure 3, the increased neuroinflammation should be quantified by flow cytometry, not immunohistochem and western blot. Many myeloid cells can express IBa1 and it does not mean that they are reactive.

Response: We have performed flow cytometry to quantify the reactive microglia. Microglia are resident brain macrophages, can be activated in SAH and experience classical proinflammatory M1 phenotype and alternative anti-inflammatory M2 phenotype polarization¹. Whole brains were subjected to flow cytometry to determine the ratio of M1- to M2-like microglia. $\text{CD11b}^+ \text{CD45}^{\text{low}}$ cell populations were defined as activated microglia, and we measured the proportion of cells stained for the classical activation marker CD16/32 and the alternative activation marker CD206 in these populations. The gating strategy is shown in Fig. 4A and the representative plot graphs and quantifications are shown in Fig. 4B-D for the visudyne photoablation experiments and shown in Fig. 5D-G for the VEGFR3 tyrosine kinase inhibitor experiments.

1. Li, R. et al. TSG-6 attenuates inflammation-induced brain injury via modulation of microglial polarization in SAH rats through the SOCS3/STAT3 pathway. *J Neuroinflammation* 15, 231, doi:10.1186/s12974-018-1279-1 (2018).

5. In figure 4, the authors give the impression that ablation of lymphatics leads to worsened cognitive performance; however, no post hoc tests show a difference between L+V+SAH and V+SAH.

Response: We considered the main reason for no post hoc tests showing a difference between L+V+SAH and V+SAH is lack of sufficient sample sizes. So we increased the number of mice to 19~22 mice per group (pooled from 2 independent experiments). We observed significant difference between L+V+SAH and V+SAH in outcomes of open field test and the number of entries into the novel arm of Y-maze test (data shown in Fig. E-H).

6. I don't understand how dynamic lymphatic flow can be quantified based a static picture. This needs more explanation, ir it must be done by 2 photon microscopy.

Response:

Macromolecules in CSF are drained by meningeal lymphatics into CLNs, and the faster the lymph flow the more macromolecules transported to CLNs. Thus, we injected the antibody and fluorescence microbeads (1 μm in diameter) into CSF with the same injection speed among groups and left to flow for 15 minutes and 2 hours, respectively. The area of antibody and microbeads in dCLN was used to present the lymphatic function. The more antibody and microbeads carried to dCLNs means quicker lymph flow.

But we strongly agree that the two photon microscopy is a compelling method to measure the dynamic lymph flow. It is a pity that we do not have access to a two photon microscope in our institute. But we found that indocyanine green near-infrared (ICG-NIR) imaging has also been reported to assess lymph flow of afferent lymphatic vessels of CLNs and peripheral lymphatic vessels^{1,2}. So we used this method to quantify the contraction frequency of mandibular LN afferent lymphatic vessels. As shown in Fig. 6I-J, the contraction frequency was significantly higher in SAH group.

1. Qiaoli Ma, B. V. I., Michael Detmar, Steven T. Proulx. Outflow of cerebrospinal fluid is predominantly through lymphatic vessels and is reduced in aged mice. *Nat Commun* 8(1):1434, doi:10.1038/s41467-017-01484-6 (2017).

2. Qianqian Liang, Yawen. Ju, Yan Chen, Wensheng Wang, et al. Lymphatic endothelial cells efferent to inflamed joints produce iNOS and inhibit lymphatic vessel contraction and drainage in TNF-induced arthritis in mice. *Arthritis Research & Therapy* 18:62, doi:10.1186/s13075-016-0963-8 (2016).

Minor

There are numerous grammatical errors throughout the manuscript in tense and plurality.

Response: The revised manuscript has been edited by a native English speaker. If the reviewer observes any remaining errors please do let us know and we will fix them.

All data should be shown as mean and standard deviation, not standard error of the mean, which is much smaller than the standard deviation.

Response: We have replaced all the standard errors by standard deviations.

Reviewers' Comments:

Reviewer #1:

Remarks to the Author:

my comments have been addressed

Reviewer #2:

Remarks to the Author:

The authors provide a revised manuscript aimed to determine the relevance of the meningeal lymphatic system on the clearance of intracranial erythrocytes after subarachnoid hemorrhage. Overall the manuscript is much improved by the additional experiments.

A few remaining concerns warrant discussion:

- 1- The authors refer to the lymphatic zone of the cervical lymph nodes, but more standard terminology for lymph node anatomy is suggested.
- 2- Why is there parenchymal hemorrhage in the subarachnoid blood injection model (Fig 3G)?
- 3- The terminology of M1/M2 microglia is no longer considered straightforward in vivo, and the authors should be careful calling the cells M1 or M2 based on 2 markers. Furthermore, it appears they are quantifying single positives (CD206 or CD16/32) and ignoring the double positive gate.
- 4- The authors should not report the trends or numerical differences when there is no statistical significance (see CD206 results of MAZ51 + SAH section).

Reviewer #3:

Remarks to the Author:

The authors have appropriately addressed all of my comments. Good job.

Dear editor,

We are appreciated for the opportunity to revise our manuscript again. We have addressed all the comments, and believe these helped us improve the manuscript markedly.

Reviewer #2 (Remarks to the Author):

The authors provide a revised manuscript aimed to determine the relevance of the meningeal lymphatic system on the clearance of intracranial erythrocytes after subarachnoid hemorrhage. Overall the manuscript is much improved by the additional experiments.

A few remaining concerns warrant discussion:

1- The authors refer to the lymphatic zone of the cervical lymph nodes, but more standard terminology for lymph node anatomy is suggested.

Response: We totally agree with the suggestion, and use a more standard terminology “Lyve-1 positive lymphatic sinus” to replace “lymphatic zone” in the manuscript.

2- Why is there parenchymal hemorrhage in the subarachnoid blood injection model (Fig 3G)?

Response: This is an excellent question. Our data (Fig. 3G, mice in first 3 groups) show that the mice in SAH, L+SAH, and V+SAH group do not have blood clots on the pons and medulla, indicating that the blood clots may not come from parenchymal hemorrhage. Cerebrospinal fluid (CSF) is produced within the cerebral ventricular system and circulates from the cerebral ventricles toward the subarachnoid space. Previous studies report that CSF flow is not unidirectional, ventricular reflux of CSF from a cistern is observed in patients with hydrocephalus. Ventricular blood is presented in some acute subarachnoid hemorrhage, which is proposed that is refluxed from cisternal hemorrhage and not indicative of primary ventricle bleeding^{1,2}. Thus we considered that clots on the pons and medulla (Fig. 3G, L+V+SAH group) may be deposited by the blood in the fourth ventricle refluxed from subarachnoid blood.

Reference:

1. Park JH, Park YS, Suk JS, et al. Cerebrospinal Fluid Pathways From Cisterns to Ventricles in N-butyl Cyanoacrylate-Induced Hydrocephalic Rats. *J Neurosurg Pediatr* 8 (6), 640-6 (2011).

2. Wilson CD, Safavi-Abbasi S, Sun H, et al. Meta-analysis and Systematic Review of Risk Factors for Shunt Dependency After Aneurysmal Subarachnoid Hemorrhage. *J Neurosurg* 126 (2), 586-595 (2017).

3- The terminology of M1/M2 microglia is no longer considered straightforward in vivo, and the authors should be careful calling the cells M1 or M2 based on 2 markers. Furthermore, it appears they are quantifying single positives (CD206 or CD16/32) and ignoring the double positive gate.

Response: We totally agree with the suggestion. We replace the terminology of

M1/M2 microglia by CD16/32+CD206- pro-inflammatory and CD206+CD16/32- anti-inflammatory microglia. In addition, we have quantified the double positive cells (CD206+CD16/32+) and show in Fig 4E, 5H.

4- The authors should not report the trends or numerical differences when there is no statistical significance (see CD206 results of MAZ51 + SAH section).

Response: We appreciate the suggestion, and delete the description for no statistically significant data in the manuscript.